

SciPost Phys. Lect. Notes 4 (2018)

# Quantifying the effect of interactions
# in quantum many-body systems

**Jiannis K. Pachos[1*] and Zlatko Papić[1]**

**1** School of Physics and Astronomy, University of Leeds, Leeds, LS2 9JT, United Kingdom

⋆ j.k.pachos@leeds.ac.uk

## Abstract

Free fermion systems enjoy a privileged place in physics. With their simple structure they can explain a variety of effects, ranging from insulating and metallic behaviours to superconductivity and the integer quantum Hall effect. Interactions, e.g. in the form of Coulomb repulsion, can dramatically alter this picture by giving rise to emerging physics that may not resemble free fermions. Examples of such phenomena include high-temperature superconductivity, fractional quantum Hall effect, Kondo effect and quantum spin liquids. The non-perturbative behaviour of such systems remains a major obstacle to their theoretical understanding that could unlock further technological applications. Here, we present a pedagogical review of "interaction distance" [Nat. Commun. 8, 14926 (2017)] – a systematic method that quantifies the effect interactions can have on the energy spectrum and on the quantum correlations of generic many-body systems. In particular, the interaction distance is a diagnostic tool that identifies the emergent physics of interacting systems. We illustrate this method on the simple example of a two-site Fermi-Hubbard model.

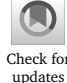

# 1  Introduction

The study of interacting quantum systems is recognised as one of the hardest problems in physics. When interactions are weak, perturbation theory and mean-field theory can be successfully employed to find an approximate description of a system. In a handful of cases, exact solutions to idealised models are also available, usually in low-dimensional systems [1]. However, a systematic approach for solving general strongly-interacting systems does not exist. To date, there has been a wide variety of inspired theoretical methods applicable to particular kinds of interacting systems, including, e.g., density functional theory [2–6], bosonisation [7], AdS/CFT correspondence [8–10], variational methods [11–13], the Bethe ansatz for integrable systems [14], or advanced numerical techniques based on quantum entanglement [15–17] and quantum Monte Carlo [18].

Physical systems are typically described by Hamiltonians which comprise a kinetic and an interaction term. Usually we associate the strength of interactions with the magnitude of the coupling constant in front of the interaction term. While this might be valid in the perturbative regime, it can also happen that models in the non-perturbative regime can be recast as free theories with a modified kinetic operator. Then, the interaction coupling fails to identify how truly interacting a system is, requiring a new measure of "interactiveness". Furthermore, it is of much interest to identify the "optimal" free theory, i.e., the theory that gives the closest possible description to the given interacting system. Finally, one would like to quantify the "error" in approximating the interacting system by the optimal free theory. Such a generally applicable diagnostic tool is still lacking.

A common approach for finding a free theory from an interacting one is by employing the mean-field approach [19]. This approach assumes weak correlations, such that the Wick theorem [20] can be approximatively applied and modifications of the kinetic term of the original Hamiltonian can be obtained. The mean-field approach is constructive in the sense it can determine the solution of the system when it is weakly correlated, but fails in general to provide the optimal free model. Powerful extensions of this approach, like the density functional theory, can identify the free fermion theory that has the same kinetic term and local fermion density as the interacting theory. Nevertheless, these methods require some additional insight into the form of the correlation and exchange functionals. Several other methods have been employed with rather specialised applicability [21–27].

Here we give an introduction to *interaction distance* [28], a systematic method that allows to quantify the effect of interactions on a generic quantum system. Interaction distance is formulated in terms of quantum information concepts, such as the entanglement spectrum [29] and distinguishability measures between quantum states [30]. As a result, interaction distance can be evaluated from the data obtained by analytic or numerical studies of entanglement or energy spectra of the system. This tool not only quantifies the effect of interaction on a quantum system, but also allows to identify the optimal free model closest to the interacting one.

For example, if the interaction distance is zero then the interacting system can be faithfully described by the optimal free system. Importantly, this can occur even in the non-perturbative regime of strong interactions. Hence, the interaction distance is a versatile tool for probing many-body systems, complementary to similar diagnostics such as the entanglement or Renyi entropies [30]. Knowing if an interacting model behaves effectively as free can also be used as a resource for scientific or technological applications, such as quantum simulations or quantum computation. Moreover, it can inspire new theoretical approaches for analytically solving strongly interacting systems.

The remainder of this paper is organised as follows. We start by reviewing some general properties of free and interacting systems and highlighting their differences in Secs. 2 and 3 . While interaction distance can be formulated for any kind of system (e.g., bosons and fermions, on a lattice or in continuum), for technical simplicity in this paper we focus on fermionic lattice models that have a finite local Hilbert space. In Sec. 4 we introduce some technical concepts, such as thermal and reduced density matrix, which will allow us to define the interaction distance. The motivation behind interaction distance is provided in Sec. 5, while its formal definition and general properties are presented in Sec. 6. In Sec. 7 we develop an intuitive understanding of interaction distance in the perturbative regime of weak interactions. Sec. 8 illustrates a simple application of interaction distance to the solvable two-site Fermi-Hubbard model. Our conclusions and future directions are presented in Sec. 9. Further examples and numerical code can be found at Interaction Distance Website [1].

## 2   Free-fermion systems

Free-fermion systems form the foundation of our understanding of the majority of physical phenomena. Due to their analytical tractability they can transparently describe many physically relevant systems, such as the structure of atoms or the electronic properties of solids. A typical example of the latter is a free spinless fermion chain with $N$ sites, as illustrated in Fig. 1. This model has a local Hilbert space $\{|0\rangle, |1\rangle\}$ at site $i$ with fermionic mode operators $c_i^\dagger$ and $c_i$, that respectively create and annihilate a fermion at site $i$, and satisfy $\{c_i, c_j^\dagger\} = \delta_{ij}$. These basis states correspond to the site being either empty, $c_i|0\rangle = 0$, or filled with one fermion, $|1\rangle = c_i^\dagger|0\rangle$. The population of each mode is an eigenvalue of $\hat{n}_i = c_i^\dagger c_i$. The full system can be described by the basis states $|n_1, n_2, \ldots, n_N\rangle$, with each $n_i = 0$ or 1. This basis spans the Fock space, which is a $2^N$-dimensional Hilbert space.

The fermions of the free system are allowed to hop between sites $i$ and $j$, which is formally implemented by the kinetic energy operator of the form $c_i^\dagger c_j + c_j^\dagger c_i$. In addition, there might be local (on-site) chemical potential, $c_i^\dagger c_i$, arising, e.g., due to the presence of an impurity at site $i$, as shown in Fig. 1. Finally, the system might also be coupled to a bath with which it can exchange particles. This might give rise to "superconducting" or "pairing" terms like $c_i c_j + c_j^\dagger c_i^\dagger$. Thus, in general the model contains any "quadratic" combinations of fermion operators ($c_i^\dagger c_j$, $c_i c_j$, and $c_i^\dagger c_j^\dagger$), but there are no higher order terms, like $c_i^\dagger c_j^\dagger c_i c_j$, which would mediate "interaction" between fermions.

The simple model in Fig. 1 already contains much interesting physics. For example, it can explain conductivity of metals because fermion hopping gives rise to transport and Bloch bands [31]. On the other hand, if the chemical potential varies strongly between different lattice sites in a random way, the system might undergo Anderson localisation and the transport would vanish [32]. Further, in some parameter regimes, the model can represent a one-dimensional $p$-wave superconductor, which displays a special type of boundary excitations –

---

[1]http://theory.leeds.ac.uk/interaction-distance

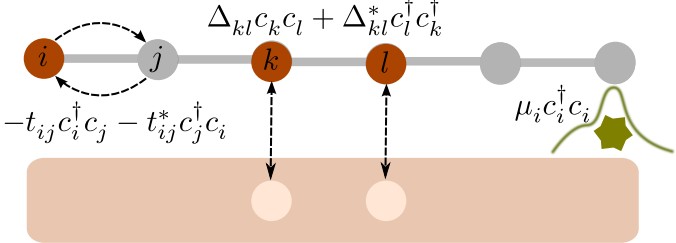

Figure 1: An example of a free fermion system defined on a one-dimensional chain with sites $i$, $j$, $k$, $l$,$\cdots$. In addition to the hopping term $c_i^\dagger c_j$ between any pairs of sites $i$ and $j$, the system is also coupled to a bath with which it can exchange pairs of particles. The latter gives rise to superconducting terms of the form $c_i^\dagger c_j^\dagger$. Local chemical potential, $c_i^\dagger c_i$, may arise, e.g., due to an impurity on site $i$.

the "Majorana zero modes" that exhibit non-Abelian anyonic statistics [33]. Finally, the effects of electromagnetic fields can be described by introducing complex phases into the fermion hopping amplitudes, $t_{ij}$. In two dimensions, this can give rise to "Chern bands" [34] – lattice analogs of integer quantum Hall states.

The quadratic structure of the Hamiltonian allows one to directly diagonalise it using elementary techniques. For example, in the absence of superconducting terms, a lattice system with $N$ sites is described by a general Hamiltonian

$$H = \sum_{i,j=1}^{N} h_{ij} c_i^\dagger c_j. \tag{1}$$

Here, $h$ is an $N \times N$ Hermitian matrix that can be diagonalised by an $N$-dimensional unitary transformation $u$. The new modes (eigenmodes) in which $H$ is diagonal are simply given by

$$\tilde{c}_j^\dagger = \sum_{i=1}^{N} c_i^\dagger u_{ij}, \tag{2}$$

which are a linear combination of the original $c_i$ modes. Then the Hamiltonian assumes the form

$$H = \sum_{j=1}^{N} \epsilon_j \tilde{c}_j^\dagger \tilde{c}_j, \tag{3}$$

where the eigenenergies are

$$\epsilon_j = (u^\dagger h u)_{jj}, \quad j = 1, 2, \ldots, N. \tag{4}$$

Hence, to diagonalise the $2^N$-dimensional Hamiltonian, $H$, we only need to diagonalise the kernel Hamiltonian $h$, with the help of $u$ which is $N$-dimensional. The diagonalisation of $h$ is an exponentially simpler problem. The free fermion system is also known as a single-particle problem as the eigenstates of each fermionic particle are independent from the populations of the other eigenmodes as they are constants of motion (see below). If the Hamiltonian $H$ also contains superconducting terms, the definition of $u$ can be generalised so that $\tilde{c}$ is a linear function not only of $c$ but also of $c^\dagger$. This method is known as the Bogoliubov transformation [35] and it results in a final expression which is analogous to Eq. (3).

The meaning of Eqs. (2)-(3) is that in the new basis of states $\tilde{c}_j^\dagger|0\rangle$ the system behaves as a set of independent, *uncorrelated* modes. The new single-particle modes $\tilde{c}_j$ are also fermionic

because the commutation properties are preserved by the unitarity of $u$, i.e., $\{\tilde{c}_i, \tilde{c}_j^\dagger\} = \delta_{ij}$. The problem can now be completely solved because, for the $N$-many original modes $c_j$, we have constructed exactly $N$ new modes $\tilde{c}_j$, whose population operators trivially commute with the Hamiltonian,

$$\left[\tilde{c}_j^\dagger \tilde{c}_j, H\right] = 0. \tag{5}$$

The new modes are thus integrals of motion, and they provide conserved quantum numbers – the populations of the new modes – given by the eigenvalues of $\tilde{c}_j^\dagger \tilde{c}_j$ for each $j$, which fully specify the eigenstates of the system.

Now we can consider a many particle free fermion system. Due to Pauli exclusion principle the available $N$ single-particle modes can be populated by 0, 1, ..., $N$ fermions in total. This results in an exponentially large Hilbert space of total dimension $2^N$. However, due to the factorisation of the Hamiltonian in Eq. (3), any property of the system can be obtained efficiently, requiring only a small number of parameters. For example, the energy spectrum of the system is given by

$$E_k^{\rm f}(\epsilon) = E_0 + \sum_{j=1}^{N} \epsilon_j n_j(k), \quad k = 1, 2, \ldots, 2^N, \tag{6}$$

where $\epsilon_j$ are the single-particle energies in Eq. (4) and $k$ labels all many-body states. The energies are expressed relative to the vacuum reference energy, $E_0$. For each state $k$, the set of numbers $\{n_j(k)\}$ specifies the occupancies of free modes $\tilde{c}_j^\dagger$. Each $n_j(k)$ must be 0 or 1 and it is an eigenvalue of the operator $\tilde{c}_j^\dagger \tilde{c}_j$. Thus, by assigning the $N$-many free energies $\epsilon_j$ to any allowed number of fermions we can generate the entire many-body energy spectrum, $\{E_k^{\rm f}(\epsilon)\}$, of the system. For example, with two free modes $\epsilon_1, \epsilon_2$, the many-body energy spectrum contains four levels given by $E_0$, $E_0 + \epsilon_1$, $E_0 + \epsilon_2$, $E_0 + \epsilon_1 + \epsilon_2$. Similarly, properties of eigenstates can also be computed efficiently due to the Wick theorem: as the Hamiltonian is quadratic, the computation of correlation functions can be reduced to evaluating only two-point correlators, $\langle c_i^\dagger c_j \rangle$, in the eigenstate of interest. Hence, the quantum correlations of the system can be exactly described in terms of these two-point correlators that increase only polynomially with the system size.

## 3 Interacting fermion systems

Although much interesting physics can be explained in terms of free fermions, there are also some basic phenomena, like thermalisation in closed systems, which cannot be described without invoking interactions (see the recent review [36]). When particles do not interact, they do not scatter, so the system is "non-ergodic" and fails to thermalise. In such systems, particles would propagate ballistically, unlike in generic thermalising systems where propagation is diffusive. Another phenomenon which crucially depends on the presence of interactions is the emergence of quasiparticles whose mutual braiding statistics is different from bosonic or fermionic, e.g., as arising in the fractional quantum Hall effect [13, 37] and frustrated magnetism [38–41]. In certain cases, such as the example shown in Fig. 1, a special symmetry (like fermion parity) can allow excitations with different types of statistical properties. An example of this are the Majorana zero modes [33], whose braiding statistics is different from ordinary fermions when they are restricted to two dimensions.

In the previous section, we have seen that a problem of $N$ non-interacting fermions can be reduced to the problem of finding a unitary transformation $u$ that maps the original fermionic

modes $c^\dagger$ to new independent modes, $\tilde{c}^\dagger$. This unitary diagonalises the kernel Hamiltonian, $h$, and thus its dimension scales linearly with the system size $N$, which allows to efficiently solve non-interacting systems. Let us now consider the case where we add an interaction term to the Hamiltonian. By "interaction" we formally mean any term in the Hamiltonian which is higher than quadratic in the fermion operators. For example, a density-density interaction can give a quartic Hamiltonian of the form

$$H = \sum_{i,j=1}^{N} (h_{ij} c_i^\dagger c_j + V_{ij} \hat{n}_i \hat{n}_j), \tag{7}$$

where $\hat{n}_i = c_i^\dagger c_i$ and $V_{ij}$ is the interaction coupling between particles at sites $i$ and $j$. In this case, in order to diagonalise the Hamiltonian (7), one can no longer diagonalise the kernel $h$ and the interaction couplings $V$ independently, as the two terms do not in general commute. Hence, in order to diagonalise the Hamiltonian a $2^N \times 2^N$ unitary matrix $U$ is needed, which acts on the full many-body Fock space of the system, i.e.,

$$E_k = (U^\dagger H U)_{kk}, \quad k = 1, 2, \ldots, 2^N. \tag{8}$$

These eigenvalues can all be independent from each other as they do not need to satisfy the simple relation (6). In rare instances, there may exist linearly many, mutually commuting operators $\mathscr{I}_j$ such that

$$\left[ \mathscr{I}_j, H \right] = 0, \tag{9}$$

which make the system integrable. In that case the problem can be solved exactly in a similar, albeit more complicated way, to the free fermion case [42].

   The eigenvalues $E_k$ of an interacting system can take arbitrary values, in contrast to the eigenvalues of a free system, that have the very specific structure (6) determined by the linearly many single particle energies $\epsilon_j$. Moreover, the eigenstates of the system cannot be given in terms of uncorrelated eigenmodes, as was the case for free systems. This complexity in the structure of interacting models makes them in general very hard to solve or to diagnose their properties. In the following section, we introduce two theoretical concepts – thermal states and the reduced density matrices of eigenstates – which we will use to diagnose the effect of interactions on the properties of generic interacting systems.

## 4   Thermal and reduced density matrices

The properties of a generic system can be described using two types of states: the thermal state, $\rho^{\text{th}}$, or the reduced density matrix, $\rho^{\text{ent}}$, corresponding to one of the system's eigenstates. Both of these are density matrices [43] and they provide complementary information about the statistical or entanglement properties of a quantum system.

   Let us first consider a system in thermal equilibrium. The state of such a system is given by the (thermal) density matrix [44]

$$\rho^{\text{th}}(\beta) = \frac{1}{Z} e^{-\beta H}, \tag{10}$$

where $Z = \text{tr}(e^{-\beta H})$ is the partition function and $T = 1/\beta$ (in units $k_B = 1$) is the thermodynamic temperature. Partition function can appear either explicitly, as a normalisation of the density matrix in Eq. (10), or it can be included in the spectrum as a constant energy $E_0 = \frac{1}{\beta} \ln Z$ which appeared in Eq. (6). If the system is free, then thermodynamic functions can be directly evaluated from Eq. (10) using the free-fermion factorisation of the energies

in Eq. (6). The thermal state corresponding to the free Hamiltonian (1) is also called "Gaussian" as the exponent is quadratic in the creation and annihilation operators (conversely, if the Hamiltonian is interacting, the thermal state is called "non-Gaussian"). The eigenvalues $\rho_k^{\text{th}}(\beta)$ of $\rho^{\text{th}}(\beta)$ are given in terms of the energy eigenvalues of the system

$$\rho_k^{\text{th}}(\beta) = \frac{1}{Z} e^{-\beta E_k}. \tag{11}$$

Note that, due to the exponential dependence, at low temperatures ($\beta E_k \gg 1$), only the low-lying energy eigenstates contribute to $\rho^{\text{th}}$.

Apart from the energy spectrum one can also consider the entanglement properties of a certain eigenstate $|\Psi_k\rangle$ of $H$. Quantum correlations are usually studied using the *reduced* density matrix [30]. To evaluate the reduced density matrix, we perform a bipartition of the system into a region $A$ and its complement $B$. We will consider spatial bipartition where $A$ is a contiguous region containing some number of lattice sites (typically half). This results in the decomposition of the Hilbert space $\mathscr{H} = \mathscr{H}_A \otimes \mathscr{H}_B$. For a certain (pure) eigenstate $|\Psi_k\rangle$, the reduced density matrix is then defined as

$$\rho^{\text{ent}} = \text{tr}_B |\Psi_k\rangle\langle\Psi_k|, \tag{12}$$

where $\text{tr}_B$ denotes the partial trace over the degrees of freedom in $B$. Reduced density matrix $\rho^{\text{ent}}$ fully characterises the state of the subsystem $A$ and in general is a mixed state.

The reduced density matrix can be equivalently expressed in the following form, which brings out the similarity with the thermal density matrix in Eq. (10):

$$\rho^{\text{ent}} = e^{-H^{\text{ent}}}. \tag{13}$$

Here we have introduced the "entanglement Hamiltonian" [45] $H^{\text{ent}}$, defined on the subsystem $A$. Thus, the reduced density matrix behaves as the thermal state of the subsystem with an effective Hamiltonian $H^{\text{ent}}$ (and at fictitious temperature, $\beta^{\text{ent}} = 1$). The eigenvalues of $\rho^{\text{ent}}$ are simply related to the "entanglement spectrum" [29] $\{E^{\text{ent}}\}$ of $H^{\text{ent}}$ via $\rho_k^{\text{ent}} = e^{-E_k^{\text{ent}}}$. We will always assume that the entanglement spectrum is normalised according to

$$\text{tr}\,\rho^{\text{ent}} = \sum_k e^{-E_k^{\text{ent}}} = 1. \tag{14}$$

The eigenvalues of $\rho^{\text{ent}}$ (and as a result the entanglement spectrum) quantify the quantum correlations between $A$ and $B$, as can be easily checked with the extreme examples of separable states or maximally entangled ones. In the case of systems with conformal invariance [46,47] or in topological phases of matter [48], the entanglement spectrum inherits some characteristics of the energy spectrum of the full system, e.g., it reveals the energy excitations at the edge of a topologically ordered system [29]. Moreover, due to the exponential relation (13), the dominant quantum correlations depend primarily on the lowest part of the entanglement spectrum.

At low enough temperatures, the structure of the energy spectrum and the entanglement spectrum is often "simpler" than in a generic many-body system, especially if some quantum ordering takes place. The properties of the system can then be described in terms of effective quasiparticles, which are localised excitations that determine the dominant quantum correlations in the low-lying states of the system. As pointed out by Li and Haldane [29], the entanglement spectrum exhibits a generic separation into the universal long-wavelength part and a non-universal short-distance part, the two being separated by the "entanglement gap" [29]. Assuming that the linear size of the system's quasiparticles, $\ell$, is much smaller than the linear size of the partition $A$, the long-wavelength physics of the system is determined by correlated

quasiparticle excitations across the entanglement partition. The lengthscale $\ell$ is a function of microscopic details of the Hamiltonian, in particular it may diverge at the phase transition where the order is destroyed. In this paper we focus on the low energy or the long wavelength limit, which corresponds to probing the correlations between the quasiparticles rather than their internal structure.

## 5   From interacting to free

It is well known that even if the Hamiltonian contains interaction terms, the system may still be described by a free-electron model, exactly or approximately. This may be especially the case when one focuses on the low-energy properties. It can be argued, using adiabatic continuity, that the properties of the ground state and low-lying excitations smoothly evolve upon "switching on" interactions between particles. Hence, their fundamental properties and symmetries remain the same as without the interactions. A particularly striking example of the success of this approach is the Landau Fermi liquid theory [49], which accurately describes properties of materials (often with extremely complicated microscopic Hamiltonians) in terms of effective free fermions with renormalised parameters e.g., the mass. In such cases, it is intuitively clear that we have an emergent free-fermion description of the system, though possibly restricted to low energies. Another example is the superconducting system, where the effective pairing terms emerge from attractive interactions that cause the fermions to pair and condense. After the condensation, the interaction term can be effectively written as a pairing term, as shown in Fig. 1.

We can formally define the emergent freedom by the system having a constrained energy spectrum of the form given in Eq. (6). More precisely, we say that the system is "free" if there exists a unitary $U$ such that the eigenenergies of $U^\dagger H U$ obey Eq. (6). This definition naturally generalises our discussion in Sec. 2. In that section we employed a unitary $u$ that generated a linear transformation of the fermionic modes on the level of a single particle, as seen in Eq. (2). More generally, we now allow for the possibility that the system is free when its spectrum satisfies Eq. (6) under some *non-linear* (non-Gaussian) unitary transformation $U$ of fermionic modes in the many-body Hilbert space, as the one in Eq. (8).

Let us look at explicitly how a Hamiltonian that might be non-quadratic with respect to some fermionic operators, could be transformed into a quadratic form with respect to some other modes and vice versa. Consider a Hamiltonian of the form

$$H = \sum_{i,j} h_{ij} f_i(c)^\dagger f_j(c), \tag{15}$$

where the operators $f_j$ are defined by

$$f_j(c) \equiv U c_j U^\dagger \tag{16}$$

(and similarly for $f^\dagger$). Here, $U$ is a $2^N \times 2^N$ unitary defined on the many-body Hilbert space, and the single-particle operators $c_j^\dagger$, $c_j$ are extended to the same space by their action on Fock states $|n_1, \ldots, n_j, \ldots, n_N\rangle$, which includes the fermion anticommutation sign, $(-1)^{\sum_{i<j} c_i^\dagger c_i}$. In general, the operators $f$ are not linearly related to $c$'s. Thus, if we look at $H$ expressed in terms of $c$ operators, it will have the form of an interacting Hamiltonian. But from Eq. (15) we know that $H$ is free in terms of the $f$ operators. Hence, given an interacting Hamiltonian expressed in terms of $\{c_j\}$ operators, our goal is to find the effective $\{f_j\}$ operators and the kernel Hamiltonian $h$, which would allow us to dramatically simplify our problem.

Apart from the emergent freedom of the energy spectrum, we also introduce a weaker notion of freedom that applies only to a given eigenstate, $|\Psi_k\rangle$. Intuitively, we call a state

free if it is "close" to a Gaussian state, i.e., if the quantum correlations in its large subsystems (i.e., for subsystems $A$, $B$ whose linear dimensions are larger than the correlation length, $\xi$, and the size of the quasiparticles, $\ell$) can be approximately generated by some free fermion modes. Formally, we can express this as a condition that the entanglement Hamiltonian of the state $|\Psi_k\rangle$ is the Hamiltonian of a free system restricted to region $A$. For example, for the free lattice system in Fig. 1 and the biparition into equal halves, $H^{\text{ent}}$ is a $2^{N/2} \times 2^{N/2}$ matrix whose eigenvalues can be fully determined from $N/2$ free energies via the combinatorial formula in Eq. (6). Thus, the spectrum of the density matrix, either thermal or reduced, exhibits a special structure in free systems. In the following section, we shall employ the energy and the entanglement spectra to define a distance that measures how far the thermodynamic and correlation properties of a many-body system are from the optimal free system.

# 6  Interaction distance

In this section, we present a distance measure that quantifies how far a quantum system (or one of its eigenstates) is from the corresponding free system (or free state). This measure was first introduced in Ref. [28], where it was called "interaction distance". We first give a brief motivation and the definition of interaction distance, then discuss how it can be efficiently evaluated, and finally list some of its general properties.

## 6.1  Definition

As announced in Sec. 5, in order to analyse the properties of an interacting fermionic system with the Hamiltonian $H$, we focus on two main aspects: the distribution of its energy spectrum and of its entanglement spectrum. In particular, we compare these spectra to the corresponding spectra of free systems. To systematically determine possible deviations the interactions can bring to a system compared to the free-fermion behaviour, we would like to have a distance function between their states. To probe the energy spectrum we choose for the state to be the thermal density matrix of the system. To probe the entanglement spectrum we can choose the reduced density matrix of an eigenstate, typically of the ground state, over some bipartition.

As a first step, we define the manifold of all free or Gaussian fermion states $\mathscr{F}$. These states are given by density matrix $\sigma$ of the form

$$\sigma = \exp\left(-\beta E_0 - \beta \sum_j \epsilon_j a_j^\dagger a_j\right), \tag{17}$$

where $a_j$ are arbitrary fermion operators and $E_0$ is a $\beta$-dependent normalisation constant that ensures $\sigma$ satisfies Eq. (14). We will use the label $\sigma$ interchangeably for either a free thermal density matrix, as in Eq. (10), or a free reduced density matrix as in Eq. (13). Importantly, in both cases the energy or entanglement spectrum of $\sigma$ obeys Eq. (6). If $\sigma$ is a reduced density matrix, then $\beta = 1$ in Eq. (17) and $a_j$ are defined on a subsystem $A$. Note that $a_j$ are not necessarily equal to the original fermionic operators, $c_j$, that appear in the Hamiltonian $H$ of the system.

Next, we consider an arbitrary density matrix, $\rho$, either coming from a thermal state or from partial trace over a pure eigenstate of the system. We want to quantify if $\rho$ has the structure similar to Eq. (17), or in other words we want to determine $\sigma \in \mathscr{F}$ which is the optimal free state, i.e., "closest" to $\rho$, see Fig. 2. As emphasised above, this optimal free state does not need to be the same as the free state obtained by simply removing the interaction terms from the Hamiltonian. Hence, this approach provides a new perspective into the properties of interacting systems: it identifies how "interacting" these systems are with respect to *any*

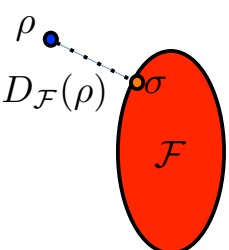

Figure 2: The manifold $\mathcal{F}$ of free states and the state $\rho$ of an interacting system that in general sits outside $\mathcal{F}$. The interaction distance $D_{\mathcal{F}}(\rho)$ is the shortest distance between the state $\rho$ and the manifold $\mathcal{F}$. Finding this distance also determines the optimal free model $\sigma \in \mathcal{F}$ that is closest to $\rho$. In general, the optimal $\sigma$ may not be unique.

free-fermion system, and it provides the optimal free models associated to them. Note that, in general, the optimal free state may not be unique. In the examples discussed in this paper, as well as in the literature [28, 50], the optimal free state was found to be unique. Thus, for simplicity, below we will assume that optimal $\sigma$ is unique.

To quantify how similar or different an interacting system is from a free one, we introduce the interaction distance $D_{\mathcal{F}}$. This is defined as the trace distance between the density matrix $\rho$ of a generic system and the closest density matrix corresponding to a free system $\sigma$, given by

$$D_{\mathcal{F}}(\rho) = \min_{\sigma \in \mathcal{F}} D(\rho, \sigma), \tag{18}$$

where $D(\rho, \sigma) = \frac{1}{2}\mathrm{tr}\sqrt{(\rho - \sigma)^2}$ is the trace distance. This distance expresses the distinguishability of the two density matrices, $\rho$ and $\sigma$. The quantity $D_{\mathcal{F}}$ has a geometric interpretation as the distance of the density matrix $\rho$ from the manifold $\mathcal{F}$, as shown in Fig. 2. Note that the trace distance is merely one convenient choice for the definition of $D_{\mathcal{F}}$, other quantities like relative entropy [30] can equally be used.

The definition of $D_{\mathcal{F}}$ involves a potentially difficult minimisation over all $\sigma$. Nevertheless, it has been shown that the minimum of $D(\rho, \sigma)$ can be obtained simply from the spectra of $\rho$ and $\sigma$ [28, 51]. More precisely, both $\rho$ and $\sigma$ can be individually diagonalised (even though $\rho$ and $\sigma$ may not have a common eigenbasis), and the problem then reduces to finding a unitary transformation $W$ which minimises $D(\rho_{\mathrm{d}}, W^{\dagger}\sigma_{\mathrm{d}}W)$, where diagonal matrices $\rho_{\mathrm{d}}, \sigma_{\mathrm{d}}$ contain the spectra of $\rho$ and $\sigma$, respectively. It can be proven [51] that the minimum of $D(\rho_{\mathrm{d}}, W^{\dagger}\sigma_{\mathrm{d}}W)$ is achieved when $W$ is a permutation matrix which orders the entries in $\rho_{\mathrm{d}}$ and $\sigma_{\mathrm{d}}$ in the same way (e.g., from largest to smallest). This significantly simplifies the minimisation procedure of Eq. (18). Instead of optimising among all the possible variables of a density matrix with $N$ modes, that would involve $2^N \times 2^N - 1$ complex numbers, we only need to optimise among the $N$ independent parameters corresponding to single particle energies $\epsilon_j$ via Eq. (6). Hence, the interaction distance takes the form

$$D_{\mathcal{F}}(\rho) = \min_{\{\epsilon_j\}} \frac{1}{2} \sum_k \left| e^{-\beta E_k} - e^{-\beta E_k^{\mathrm{f}}(\epsilon)} \right|, \tag{19}$$

where the minimisation is with respect to the $N$-many single particle energies. As $N$ increases linearly with the system size, Eq. (19) provides the means to efficiently compute the interaction distance numerically or analytically for any state of an interacting theory whenever its energy or entanglement spectrum $\{E_k\}$ is accessible.

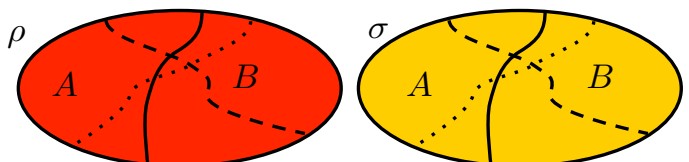

Figure 3: The eigenstate of a given interacting system (Left) can be effectively described by the eigenstate of a free one (Right). To determine that we first consider various partitions of the interacting system in $A$ and its compliment $B$ and the corresponding partitions of the free system. These partitions give rise to the reduced density matrices in $A$ denoted by $\rho$ (interacting system) and $\sigma$ (free system). If $D(\rho, \sigma) = 0$ for any of the partitions then the free system on the right is the optimal free model of the interacting system.

## 6.2 Spectral versus entanglement interaction distance

Depending on the choice of $\rho$, we can define two different types of interaction distance. For a thermal density matrix, $\rho^{\text{th}}$, we define the interaction distance for the energy spectrum, $D_{\text{th}}^{\beta}$, as

$$D_{\text{th}}^{\beta} \equiv D_{\mathcal{F}}(\rho^{\text{th}}(\beta)). \tag{20}$$

This measures the distance of the Hamiltonian spectrum, $\{E_k\}$ from the closest possible spectrum of a free fermion system, $\{E_k^{\text{f}}\}$, given by Eq. (6), through the exponentially decaying function $e^{-\beta E_k}$. As a result, this function exponentially penalises high energy eigenvalues. To compare the low energy sectors we can choose $\beta$ to be large (small temperatures), while smaller values of $\beta$ (larger temperatures) implement a comparison of larger parts of the spectrum.

The second type of interaction distance applies to an individual quantum state, defined by the reduced density matrix $\rho^{\text{ent}}$,

$$D_{\text{ent}} \equiv D_{\mathcal{F}}(\rho^{\text{ent}}). \tag{21}$$

This quantity measures the distance of the entanglement spectrum, corresponding to the given eigenstate $|\Psi_k\rangle$ and the given partition, from the closest possible free-fermion entanglement spectrum, $\{E_k^{\text{f}}\}$, given also by Eq. (6). Loosely speaking, $D_{\text{ent}}$ measures how much the part $A$ of the system "interacts" with part $B$. Similar measures to $D_{\text{ent}}$ appear in quantum information [21, 22, 27, 52–54], which however restrict to a single set of modes. The distance $D_{\text{ent}}$ can be evaluated using a formula similar to Eq. (19), with the only difference that we set the entanglement temperature to be $\beta = 1$ and the number of variational parameters $\epsilon$ is now determined by the number of degrees of freedom in the subsystem $A$.

One important difference between $D_{\text{ent}}$ and $D_{\text{th}}^{\beta}$ is that $D_{\text{ent}}$ explicitly depends on the partition between $A$ and $B$ subsystems. By varying the size of the partition $A$, in principle $D_{\text{ent}}$ can probe the short-distance physics associated with internal structure of the emerging quasiparticles rather than their correlations. In what follows, we focus on the long-wavelength limit where $D_{\text{ent}}$ is expected to have universal properties, and restrict to real space partition between contiguous regions $A$ and $B$. We will only consider as admissible partitions those for which the size of regions $A$ and $B$ is much larger than the correlation length, $\xi$ (otherwise, we would end up probing short-distance, i.e., high-energy physics, which is potentially non-universal) and the size $\ell$ of the constituent quasiparticles. Note that if we establish $D_{\text{ent}} = 0$ with respect to a certain partition, we cannot immediately conclude that correlations in $|\Psi_k\rangle$ are those of free fermions. To deduce that one needs to check that the entanglement interaction distance

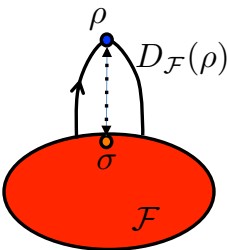

Figure 4: Possible behaviour of the density matrix $\rho$ upon increasing the interaction coupling (depicted by arrow). As the interactions are turned on, $\rho$ could depart from the free manifold $\mathcal{F}$ with an increasing interaction distance $D_{\mathcal{F}}(\rho)$. As the interaction coupling increases the interaction distance may eventually start decreasing and possibly go back to zero, thus making the system effectively free.

is zero with respect to all admissible bipartitions of the system, as shown in Fig. 3. In this case we assume that there is a global optimal Gaussian state that effectively describes the system. Nevertheless, it is possible that even if the correlations between all partitions are of free fermions, the total state is not reproducible by a Gaussian state. In the case where $D_{\text{ent}} \neq 0$ the exact value of $D_{\text{ent}}$ may then depend on the geometrical or topological properties of the partitioning boundary [50]. This is due to the possibility of different regions having different quantum correlations with their complement.

It is intriguing that a variety of distinct behaviours could emerge when we contrast the behaviour of $D_{\text{th}}$ and $D_{\text{ent}}$. For example, a system might have $D_{\text{th}} = 0$ while $D_{\text{ent}} \neq 0$ or the other way around. Moreover, $D_{\text{ent}}$ can be zero for some of the eigenstates but not for all of them. If we are interested in the low energy behaviour then we may only consider the behaviour of a few low lying states. The most interesting examples are those of interacting Hamiltonians for which $D_{\mathcal{F}} \approx 0$, where an emergent freedom arises for strong interactions, as shown in Fig. 4. In such cases, we can identify the single particle energies that reproduce the full energy or the entanglement spectrum, thus exponentially compressing the amount of information needed to describe the system.

## 6.3   General properties of interaction distance

We now describe some general properties of the thermal and entanglement interaction distances. The trace distance is bounded, i.e., it satisfies $0 \leq D(\rho, \sigma) \leq 1$ when $\rho$ and $\sigma$ are allowed to vary arbitrarily. The case $D(\rho, \sigma) = 0$ corresponds to the two density matrices being identical, while $D(\rho, \sigma) = 1$ corresponds to $\rho$ being as different from $\sigma$ as possible. Nevertheless, for determining $D_{\mathcal{F}}$ for a given $\rho$ we need to optimise with respect to $\sigma$. So it is possible that the interaction distance will not saturate the upper bound as the two density matrices are not arbitrarily chosen. Indeed, we have shown [28] that

$$0 \leq D_{\mathcal{F}} \leq 3 - 2\sqrt{2} \approx \frac{1}{6}, \tag{22}$$

so in practice the upper bound is much smaller than 1. The condition $D_{\mathcal{F}} = 0$ corresponds to a system that can be exactly described by free fermions, while $D_{\mathcal{F}} = 3 - 2\sqrt{2}$ is the maximum distance a density matrix can have from a free description. Note that even this value can be attained only for asymptotically large systems [50].

The physical meaning of $D_{\mathcal{F}}$ is that it provides a bound on the expectation values of various observables in the state $\rho$ compared to state $\sigma$. For any observable $\mathcal{O}$, this can be expressed

as

$$|\langle \mathcal{O} \rangle_\rho - \langle \mathcal{O} \rangle_\sigma| \le c_{\mathcal{O}} D_{\mathcal{F}}(\rho), \tag{23}$$

where $\langle \mathcal{O} \rangle_\rho = \text{tr}(\mathcal{O}\rho)$ and the constant $c_{\mathcal{O}}$ depends only on the norm of the operator $\mathcal{O}$, but does not depend on the states $\rho$ or $\sigma$. The ability to approximate expectation values in terms of an emerging free system signifies also the applicability of the Wick decomposition with an error which tends to zero as $D_{\mathcal{F}} \to 0$.

In terms of practical evaluation of interaction distance, we note that $D_{\text{ent}}$ can be efficiently computed by the use of Eq. (19). The required entanglement spectrum, used as input for Eq (19), can be obtained in one dimension by computationally efficient matrix-product state methods [15, 16]. The computation of $D_{\text{th}}$ is also efficient in the system size, but requires as input the energy spectrum (or its part), which is much less efficient (if a finite density of states is required, the cost would become exponential in the system size). Numerical code for evaluating $D_{\mathcal{F}}$ and the accompanying documentation can be found at [1].

Finally, we mention that there is an inherent flexibility in the definition of $D_{\mathcal{F}}$, which allows $\rho$ and $\sigma$ to be of different dimensions (possibly as a result of different statistics of their underlying modes). This flexibility is enabled by the fact that we can freely change the dimension of $\rho$ by adding (or removing) zero eigenvalues. In the case of thermal states, adding large energies does not affect the lower temperature properties; for reduced density matrices, adding zeros corresponds to adding disentangled parts in the system that do not alter its correlation properties. Hence, adding zero eigenvalues in $\rho$ does not alter the physical properties, which allows us to investigate the important cases where an emergent description of a system is in terms of different types of free quasiparticles. For example, the low-energy properties and correlations in a fermionic system may behave as bosons, rather than fermions. A simple example is the 1D XY spin model, which can be expressed as a fermion lattice model like in Fig. 1. For such a model, the correlations in the ground state would be formally given by a fermionic reduced density matrix $\rho$, while the effective $\sigma$ should be more naturally given in terms of bosonic modes. In order to compare such $\rho$ and $\sigma$ we would need to pad $\rho$ with some zeros because the Hilbert space of $\sigma$ has higher dimension (since it is a bosonic Hilbert space). The flexibility of $D_{\mathcal{F}}$ that allows to compare density matrices coming from different kinds of particles allows for a study of a much wider class of emergent phenomena in many-body systems. Technically, this can be done because we only require the spectra of $\rho$ and $\sigma$ to determine $D_{\mathcal{F}}$, as discussed in Sec. 6.1.

# 7 Perturbative analysis of interaction distance

To obtain a better understanding of the interaction distance, we now focus on the perturbative regime of interactions. For that we consider a Hamiltonian $H_0$ of $N$ free fermionic modes, where we add the fermion interactions $\hat{V}$ as a perturbation for weak dimensionless coupling $\lambda$. To simplify the analysis we assume that the free system, $H_0$, has non-degenerate eigenstates. Moreover, we assume that the energy gaps separating the adjacent eigenstates in the spectrum remain larger than $\lambda$ as the perturbation is continuously introduced from $\lambda = 0$ to a small but non-zero value. These assumptions are clearly very stringent, e.g., in a many-body system there might be large degeneracies in the excited spectrum where the typical energy spacing is $\sim 2^{-N}$, thus a perturbation by finite $\lambda$ can couple many states at once. However, in small systems such as the two-site Fermi-Hubbard model (which we solve exactly in Sec. 8) we will find that the perturbative treatment on $D_{\mathcal{F}}$ gives good agreement with exact results in the regime of weak interactions.

By perturbative analysis of the eigenvalues and eigenstates of the interacting Hamiltonian

$$H = H_0 + \lambda \hat{V}, \tag{24}$$

we can calculate the interaction distance for the energy spectrum with small coupling $\lambda$. We first consider the eigenstates $|\Psi_k^{(0)}\rangle$ and eigenvalues $E_k^{(0)}$ of the free Hamiltonian $H_0$ satisfying

$$H_0 |\Psi_k^{(0)}\rangle = E_k^{(0)} |\Psi_k^{(0)}\rangle, \tag{25}$$

for $k = 1, ..., 2^N$. As the Hamiltonian $H_0$ describes free fermions its eigenstates are given by

$$|\Psi_k^{(0)}\rangle = |n_1(k), n_2(k), ..., n_N(k)\rangle, \tag{26}$$

where $\{n_j(k), j = 1, ..., N\}$ is the population pattern of the eigenmodes corresponding to the Gaussian state $|\Psi_k^{(0)}\rangle$. The eigenmodes are unitarily related to the initial modes of the system. Similarly, the eigenvalues of $H_0$ are given by

$$E_k^{(0)} = \sum_{j=1}^{N} \epsilon_j n_j(k), \tag{27}$$

where $\{\epsilon_j, j = 1, ..., N\}$ are the single particle energies of the eigenmodes. As the perturbative corrections will change the values of the energy, we set $E_0 = 0$ in Eq. (27), but we explicitly normalise the density matrix by the partition function.

To first order in perturbation theory the eigenvalues of $H$ are given by

$$E_k = E_k^{(0)} + \lambda \langle \Psi_k^{(0)}| \hat{V} |\Psi_k^{(0)}\rangle + \mathcal{O}(\lambda^2) \tag{28}$$

and its eigenstates are given by

$$|\Psi_k\rangle = |\Psi_k^{(0)}\rangle + \lambda \sum_{m \neq k} \frac{\langle \Psi_m^{(0)}| \hat{V} |\Psi_k^{(0)}\rangle}{E_k^{(0)} - E_m^{(0)}} |\Psi_m^{(0)}\rangle + \mathcal{O}(\lambda^2). \tag{29}$$

We assumed that the resulting energies $E_k$ do not become degenerate as $\lambda$ is kept small. Hence, the optimal free model corresponding to the interacting theory for a small $\lambda$ will have the same population pattern $\{n_j(k), j = 1, ..., N\}$ as the free model $H_0$. We shall see in the following that this condition simplifies the calculation of the spectral interaction distance, as it is not necessary to apply the minimisation to identify the optimal free model.

When $\lambda = 0$ then the optimal free model of $H$ is identical to $H_0$ itself. The first order effect in perturbation theory of the energy eigenvalues, given in Eq. (28), can change the single particle energies $\epsilon_j$, defined in Eq. (27), to the new ones $\tilde{\epsilon}_j$ that correspond to the modified optimal free model. Moreover, they can have a contribution, $\Delta E_k$, that cannot be absorbed by single particle energies. Hence we can write

$$E_k = \sum_j \tilde{\epsilon}_j n_j(k) + \Delta E_k. \tag{30}$$

In view of Eq. (28) we then have

$$\sum_{j=1}^{N} (\tilde{\epsilon}_j - \epsilon_j) n_j(k) + \Delta E_k = \lambda \langle \Psi_k^{(0)}| \hat{V} |\Psi_k^{(0)}\rangle. \tag{31}$$

We now split the set of eigenstates $\{|\Psi_k^{(0)}\rangle\}$ into the part with population patterns $\{n_j(k), j = 1, ..., N\}$ with only a single population at $j = k$ and the rest of the states that

have either zero or more than one total population. There are $N$ such single occupation states that we call $|\Psi_k^{(0)}\rangle$ for $k = 1, ..., N$. For these states we take $\Delta E_k = 0$ as interactions take place only between two or more particles. Moreover, for these single occupancy states all $n_j(k)$ in Eq. (31) are zero except from one with the single occupancy, $n_k(k) = 1$ so we obtain

$$\tilde{\epsilon}_k = \epsilon_k + \lambda \langle \Psi_k^{(0)}| \hat{V} |\Psi_k^{(0)}\rangle, \quad k = 1, ..., N. \tag{32}$$

From these $N$ equations we obtain all $N$ single particle energies $\tilde{\epsilon}_j$ for $j = 1, ..., N$ that completely determine the spectrum of the optimal free model.

After determining the effect of the perturbation on the single particle energies we can go ahead and determine the purely interacting contribution of the perturbation to the eigenvalues of the energy. From the total of $2^N$ equations of Eq. (31) we used $N$ of them to determine the single particle energies so there are $2^N - N$ of them left to determine the $2^N$ parameters $\Delta E_k$. Nevertheless, $N$ of them for $k = 1, ..., N$ that correspond to single particle occupations are set to zero, so we have an exact match of equations and unknowns. Note also that when all the populations of the eigenmodes are zero $n_j(k) = 0$ for all $j = 1, ..., N$, corresponding to the $k = 0$ eigenstate then we have $\langle \Psi_0^{(0)}| \hat{V} |\Psi_0^{(0)}\rangle = 0$ as the interactions cannot be witnessed without particles, and hence $\Delta E_0 = 0$.

By solving these linear equations we finally determine both the optimal single particle energies, $\tilde{\epsilon}_j$, and the genuine interacting effect, $\Delta E_k$. We are now in position to evaluate the spectral interaction distance

$$D_{\text{th}}^\beta = \min_{\{\tilde{\epsilon}\}} \frac{1}{2} \sum_k \left| \frac{e^{-E_k\beta}}{Z} - \frac{e^{-E_k^{\text{f}}\beta}}{Z_{\text{f}}} \right| \approx \frac{1}{2} \sum_k \frac{e^{-E_k^{\text{f}}\beta}}{Z_{\text{f}}} \left| \Delta E_k \beta - \sum_l \frac{e^{-E_l^{\text{f}}\beta}}{Z_{\text{f}}} \Delta E_l \beta \right| + + \mathcal{O}((\Delta E)^2), \tag{33}$$

where $E_k^{\text{f}} = \sum_j \tilde{\epsilon}_j n_j(k)$, with $\tilde{\epsilon}_j$ determined from Eq. (32). We have used the fact that

$$e^{-E_k\beta} \approx e^{-E_k^{\text{f}}\beta}(1 - \beta \Delta E_k), \tag{34}$$

$$Z \approx Z_{\text{f}} - \sum_l e^{-\beta E_l^{\text{f}}} \beta \Delta E_l. \tag{35}$$

Note, that no optimisation was needed in obtaining this analytic expression for the spectral interaction distance, $D_{\text{th}}^\beta$.

Similar calculation in first order perturbation can be performed for the entanglement interaction distance, but it is more cumbersome to express the final result in closed form. In the following section, we consider the two-site Fermi-Hubbard model. For this model, we analytically evaluate both spectral and entanglement interaction distance and compare them against numerical optimisation, finding excellent agreement in the regime where first order perturbation theory is applicable.

## 8 Example: Two-site Fermi-Hubbard model

We now consider the Fermi-Hubbard model on a lattice with two sites, also known as the "Hubbard dimer" [55, 56]. This model will illustrate how to analytically evaluate the interaction distance, and compare the exact results with perturbation theory. At the same time, this simple toy model is physically interesting because it features an analogue to the Mott metal-insulator transition, and it has recently been experimentally realised using cold atoms [57].

The two-site Fermi-Hubbard model is described by the following second-quantised Hamiltonian

$$
\begin{aligned}
\hat{H} &= -t\left(c_{1,\uparrow}^{\dagger}c_{2,\uparrow} + c_{2,\uparrow}^{\dagger}c_{1,\uparrow}\right) - t\left(c_{1,\downarrow}^{\dagger}c_{2,\downarrow} + c_{2,\downarrow}^{\dagger}c_{1,\downarrow}\right) \\
&\quad + \Delta_1\left(\hat{n}_{1,\uparrow} + \hat{n}_{1,\downarrow}\right) + \Delta_2\left(\hat{n}_{2,\uparrow} + \hat{n}_{2,\downarrow}\right) + \hat{V}.
\end{aligned}
\tag{36}
$$

Here $t$ denotes the hopping strength (which is set to $t = 1$) and $\Delta_j$ denotes the on-site potential (which is also fixed to $\Delta_1 = -\Delta_2 = 1$). Fermions interact with strength $V$ when they are on the same lattice site

$$
\hat{V} = V\left(\hat{n}_{1,\uparrow}\hat{n}_{1,\downarrow} + \hat{n}_{2,\uparrow}\hat{n}_{2,\downarrow}\right).
\tag{37}
$$

The Hamiltonian $\hat{H}$ in Eq. (36) commutes with the total projection of spin $\hat{S}_z = \frac{1}{2}\sum_j(\hat{n}_{j,\uparrow} - \hat{n}_{j,\downarrow})$, hence we can restrict the problem to the largest sector with $S_z = 0$, which contains four states: $|0X\rangle, |\uparrow\downarrow\rangle, |\downarrow\uparrow\rangle, |X0\rangle$, where $X$ denotes double occupancy of a given site.

In the absence of interactions ($V = 0$), the energy eigenvalues are $E_1^{(0)} = -2\sqrt{2}$, $E_2^{(0)} = E_{2'}^{(0)} = 0$ and $E_3^{(0)} = 2\sqrt{2}$, with the corresponding (unnormalised) eigenstates denoted by $|1\rangle, |2\rangle, |2'\rangle, |3\rangle$:

$$
\begin{aligned}
|1\rangle &= (3 + 2\sqrt{2})|0X\rangle + (-1 - \sqrt{2})|\downarrow\uparrow\rangle + (1 + \sqrt{2})|\uparrow\downarrow\rangle + |X0\rangle, \\
|2\rangle &= |\uparrow,\downarrow\rangle + |\downarrow\uparrow\rangle, \tag{38} \\
|2'\rangle &= |0X\rangle + |\downarrow\uparrow\rangle - |\uparrow\downarrow\rangle - |X0\rangle, \tag{39} \\
|3\rangle &= (3 - 2\sqrt{2})|0X\rangle + (-1 + \sqrt{2})|\downarrow\uparrow\rangle + (1 - \sqrt{2})|\uparrow\downarrow\rangle + |X0\rangle.
\end{aligned}
$$

Choosing the lowest energy as the reference (vacuum) energy, the single particle energies of the effective free system are

$$
\epsilon_1 = \epsilon_2 = 2\sqrt{2}.
\tag{40}
$$

According to Eq. (32), the modified single-particle energies in first order perturbation are given by

$$
\begin{aligned}
\tilde{\epsilon}_1 &= \epsilon_1 + \langle 2|\hat{V}|2\rangle = 2\sqrt{2}, \tag{41} \\
\tilde{\epsilon}_2 &= \epsilon_2 + \langle 2'|\hat{V}|2'\rangle = 2\sqrt{2} + \frac{V}{2}. \tag{42}
\end{aligned}
$$

As the interaction term does not couple the degenerate eigenstates, i.e., $\langle 2|\hat{V}|2'\rangle = 0$, Eqs. (41)-(42) can be directly applied. Further, note that the vacuum energy is also renormalised by interactions

$$
E_1 = E_1^{(0)} + \langle 1|\hat{V}|1\rangle = -2\sqrt{2} + \frac{3V}{4}.
$$

Hence, with respect to the new vacuum, the effective free particle energies are

$$
\begin{aligned}
\tilde{\epsilon}_1 &= 2\sqrt{2} - \frac{3V}{4}, \tag{43} \\
\tilde{\epsilon}_2 &= 2\sqrt{2} - \frac{V}{4}. \tag{44}
\end{aligned}
$$

The final expression for the interaction distance via Eq. (33) (setting $\beta = 1$) is

$$
\begin{aligned}
D_{\text{th}}^{\beta} &= \frac{1}{2}\left(\left|\frac{e^{-\beta(-2\sqrt{2} + \frac{3V}{4})}}{Z} - \frac{1}{Z_{\text{f}}}\right| + \left|\frac{1}{Z} - \frac{e^{-\beta\tilde{\epsilon}_1}}{Z_{\text{f}}}\right|\right. \\
&\quad \left. + \left|\frac{e^{-\beta\frac{V}{2}}}{Z} - \frac{e^{-\beta\tilde{\epsilon}_2}}{Z_{\text{f}}}\right| + \left|\frac{e^{-\beta(2\sqrt{2} + \frac{3V}{4})}}{Z} - \frac{e^{-\beta\tilde{\epsilon}_1 - \beta\tilde{\epsilon}_2}}{Z_{\text{f}}}\right|\right),
\end{aligned}
\tag{45}
$$

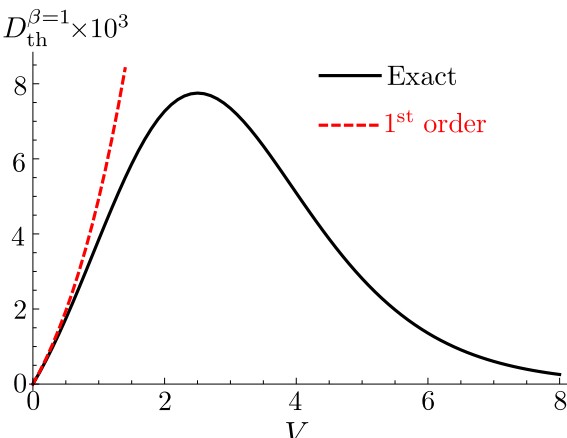

Figure 5: Interaction distance $D_{th}^{\beta}$ for the energy spectrum as a function of interaction strength $V$ in the Hubbard dimer for $\beta = 1$. Solid line denotes the exact result with $D_{th}^{\beta}$ determined by numerical optimisation. The first order perturbation in $V$, of $D_{th}^{\beta}$ according to Eq. (45) (red dashed line) shows good agreement with the exact result for weak $V \lesssim 1$. We observe that $D_{th}^{\beta}$ becomes maximum near the crossing point $V \approx 2$, where the energy gap is smallest, thus allowing the interactions to have a strong effect on the distribution of the energy eigenvalues. The interaction distance, $D_{th}^{\beta}$, tends to zero for large $V$, signalling that the model becomes effectively free, which is a non-perturbative result.

where the corresponding partition functions are

$$Z = \exp\left(-\beta(-2\sqrt{2} + \frac{3V}{4})\right) + 1 + \exp\left(-\beta\frac{V}{2}\right) + \exp\left(-\beta(2\sqrt{2} + \frac{3V}{4})\right), \quad (46)$$

$$Z_f = 1 + \exp\left(-\beta\tilde{\epsilon}_1\right) + \exp\left(-\beta\tilde{\epsilon}_2\right) + \exp\left(-\beta\tilde{\epsilon}_1 - \beta\tilde{\epsilon}_2\right). \quad (47)$$

In Fig. 5 we compare the first-order perturbation formula, Eq. (45), against the exact result (solid curve) where the full $D_{th}$ optimisation is performed numerically. We fix $\beta = 1$ in Fig. 5. The perturbative result in Eq. (45) (red dashed line in Fig. 5) shows good agreement with the full calculation at small values of $V \lesssim 0.5$. Perturbative result, as expected, significantly deviates from the full calculation in the vicinity of the transition ($V \gtrsim 2$).

It is instructive to consider also $D_{th}^{\beta}$ as a function of temperature, $T = 1/\beta$, and interaction coupling $V$, as shown in Fig. 6. We note that for small temperatures (large $\beta$) and for large temperatures (small $\beta$) $D_{th}^{\beta}$ is almost zero for all $V$. Indeed, for small temperatures only a few energy levels contribute significantly in $D_{th}^{\beta}$, for which it is possible to find a free model to approximate them. For temperatures dictated by the interaction coupling $\beta \sim 1/V_c$, where $V_c \sim 2.5$ determines the critical coupling that causes the transition of the dimer, the interaction distance becomes maximum. Hence, this result suggests that $D_{th}^{\beta}$ can diagnose the strong effect of the interactions when the system is close to its critical point. We also observe that $D_{th}^{\beta}$ becomes zero for large temperatures as well (small $\beta$). In that case, the eigenvalues of the density matrix become roughly equal, and because there are four of them, it is possible to find an effective free model. Moreover, it is apparent from Figs. 5 and 6 that the model becomes free when $V$ is small or very large. The small coupling optimal free model corresponds to the free spin-1/2 fermion system given by $V = 0$. The Bethe ansatz analysis of the large $V$ limit gives that the model is faithfully described by a free spinless fermion model with twisted boundary conditions [58].

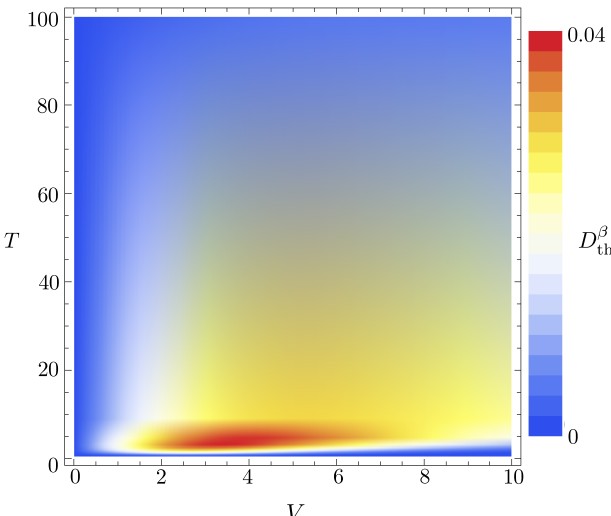

Figure 6: Interaction distance $D_{\text{th}}^{\beta}$ (colour scale) for the energy spectrum as a function of interaction strength $V$ and temperature $T = 1/\beta$ in the two-site Fermi-Hubbard model.

Finally, we consider the entanglement interaction distance of the Fermi-Hubbard dimer when it is partitioned in the middle. Perturbative analysis of $D_{\text{ent}}$ in the limit of weak interactions can be performed analogously to the energy spectrum. Since the ground state of the dimer is unique, it is sufficient to use non-degenerate first order perturbation. When $V = 0$, the eigenvalues of the reduced density matrix $\rho_A$ are given by

$$\rho_1^{(0)} = \frac{3+\sqrt{2}}{8}, \quad \rho_2^{(0)} = \rho_3^{(0)} = \frac{1}{8}, \quad \rho_4^{(0)} = \frac{3-2\sqrt{2}}{8}. \tag{48}$$

This gives us the effective free-particle "entanglement energies"

$$\epsilon_1 = \epsilon_2 = \ln(3 + 2\sqrt{2}). \tag{49}$$

From the perturbed ground state, we can determine the perturbed eigenvalues of $\rho_A$. Keeping corrections up to linear order $\mathcal{O}(V)$, we get

$$\begin{aligned}
\tilde{\rho}_1 &= \rho_1^{(0)} + \frac{(-8-5\sqrt{2})V}{128}, \quad \tilde{\rho}_2 = \rho_2^{(0)} + \frac{10\sqrt{2}V}{128}, \\
\tilde{\rho}_3 &= \tilde{\rho}_2, \quad \tilde{\rho}_4 = \rho_4^{(0)} + \frac{(8-5\sqrt{2})V}{128},
\end{aligned} \tag{50}$$

where, conveniently, we still have $\sum_k \tilde{\rho}_k = 1$. Eq. (50) is valid for weak interactions such that the ordering of the levels remains unchanged, i.e., $\tilde{\rho}_1 > \tilde{\rho}_2 = \tilde{\rho}_3 > \tilde{\rho}_4$. Assuming $V$ is small, we see that the effective single particle energies are going to be

$$\tilde{\epsilon}_1 = \tilde{\epsilon}_2 = \ln\frac{\tilde{\rho}_1}{\tilde{\rho}_2} = \ln\left(\frac{48 + 32\sqrt{2} + (-8 - 5\sqrt{2})V}{16 + 10\sqrt{2}V}\right), \tag{51}$$

which takes into account the renormalisation of the "vacuum" energy due to the perturbation. Failing to include the renormalisation of the reference energy would result in $D_{\text{ent}}$ that depends linearly on $V$, which does not capture the qualitative behaviour of the exact solution.

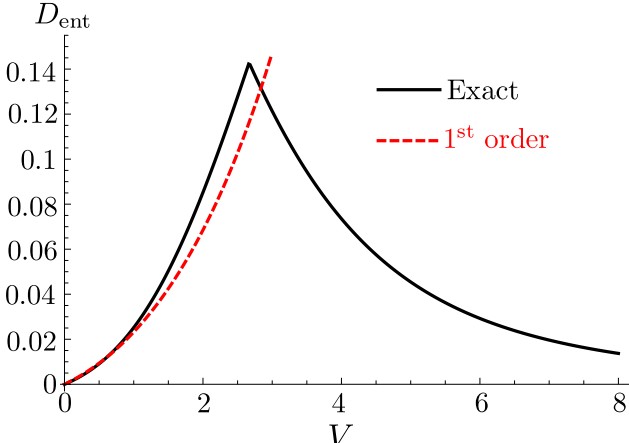

Figure 7: Interaction distance, $D_{\text{ent}}$, evaluated from the entanglement spectrum of the ground state of the two-site Fermi-Hubbard model, as a function of interaction strength $V$. The perturbative result (red dashed line) faithfully approximates the exact numerical result (black solid line) for $V \lesssim 1$. The ground state for large $V$ becomes effectively Gaussian again, as diagnosed by $D_{\text{ent}} \to 0$.

With the perturbed free-particle entanglement energies in Eq. (51), the expression for $D_{\text{ent}}$ is formally very similar to that of $D_{\text{th}}$ in Eq. (45):

$$D_{\text{ent}} = \frac{1}{2}\left( \left| \tilde{\rho}_1 - \frac{1}{Z_{\text{f}}} \right| + \left| \tilde{\rho}_2 - \frac{e^{-\tilde{\epsilon}_1}}{Z_{\text{f}}} \right| + \left| \tilde{\rho}_3 - \frac{e^{-\tilde{\epsilon}_2}}{Z_{\text{f}}} \right| + \left| \tilde{\rho}_4 - \frac{e^{-\tilde{\epsilon}_1 - \tilde{\epsilon}_2}}{Z_{\text{f}}} \right| \right), \tag{52}$$

where the free partition function $Z_{\text{f}}$ is given by the previous Eq. (47) with new $\tilde{\epsilon}_i$ defined in Eq. (51).

Fig. 7 shows the value of $D_{\text{ent}}$ as a function of $V$, for fixed $t = 1$ and $\Delta_1 = -\Delta_2 = 1$ [56]. Entanglement interaction distance behaves similarly to the spectral interaction distance $D_{\text{th}}$ except that it displays a much sharper kink around the expected thermodynamic-limit transition point, $V \approx 2.5$. The first-order perturbation result in Eq. (52) is plotted by red dashed line in Fig. 7. Similar to the interaction distance for the energy spectrum, $D_{\text{ent}}$ shows excellent agreement with the exact result for weak interactions $V \lesssim 1$. For larger $V$, the perturbation theory fails. Nevertheless, $D_{\text{ent}}$ goes to zero as $V$ increases. This signifies that correlation properties of the ground state can be described by a free theory [58].

## 9 Conclusions and outlook

The interaction distance, $D_{\mathcal{F}}$, provides a systematic method to quantify how interacting a system is with respect to its spectral and quantum correlation properties. Identifying if an interacting system behaves as free is equivalent to finding out if it satisfies the simplest possible integrability condition, that of a free system. Equivalently, it determines if a free theory can effectively describe the low energy sector of the model. When $D_{\mathcal{F}}$ approaches zero then, at least, the low energy part of the system can be efficiently described by free fermions, possibly different from the free fermions that describe the model when the interactions are turned off.

Determining the interaction distance, $D_{\mathcal{F}}$, of a state $\rho$ also specifies the optimal free state $\sigma$ that is closest to $\rho$. In principle, the parent Hamiltonian – the optimal free model – that gives rise to $\sigma$ can also be obtained, although this is in general exponentially more difficult than just determining the value of $D_{\mathcal{F}}$. The resulting free model is optimal either with respect

to the spectral or correlation properties of the interacting system. As such it is bound to behave better than the usual techniques employed to describe interacting systems, such as the mean-field theory. In particular, mean-field theory constructs a free model by using the same fermionic operators $\{c_j\}$ (or their linear combinations) as the interacting theory. This should be contrasted to the generality of our approach where optimisation over arbitrary rotations of $\{c_j\}$ in the many-body Fock space is performed (i.e., allowing for *non-linear* transformations of $\{c_j\}$). Nevertheless, it should be emphasised that, unlike mean-field theory, our method is a diagnostic tool. To apply our approach, one first needs to find the spectrum of the Hamiltonian or its ground state, before determining $D_{\mathscr{F}}$, similar to other diagnostic tools such as the entanglement entropy or finite-size scaling analysis.

While the perturbative behaviour of the interaction distance follows the intuitive expectations, as we have seen in Section 8, the non-perturbative behaviour of interacting models can be rather surprising. Often, a system appears to be free even in the presence of strong interactions when the thermodynamic limit is taken. As an example, the Ising model in both transverse and longitudinal fields [28] appears to have zero interaction distance when both field strengths are comparable in magnitude to the nearest neighbour coupling. Another, even more surprising example, is the Fermi-Hubbard model. The ground state of the 1D Fermi-Hubbard model appears to have zero entanglement interaction distance for all values of the interaction coupling when the thermodynamic limit is approached with system sizes $N = 4k + 2$, $k = 0, 1, \dots$ [56]. This generalises previous results about the ability to describe the Fermi-Hubbard model by free fermions when its interaction coupling is infinite.

When considering a system that undergoes a second-order phase transition, its critical region may be particularly susceptible to interactions. In this region the energy gap that protects the eigenstates from perturbations becomes negligibly small (and the correlation length diverges), thus exposing them to interactions. The scaling analysis of $D_{\text{ent}}$ near criticality for various systems sizes reveals important universal features about the systems. Most importantly, it reveals if the interaction term is a relevant or irrelevant operator in the renormalisation group sense [28]. An important open problem is to relate the scaling exponents of $D_{\text{ent}}$ to the underlying critical theory (e.g., conformal field theory in one-dimensional cases).

Away from critical regions we can consider gapped systems that are fixed points of renormalisation group. Among these systems, of particular interest are those that support topologically non-trivial properties. Examples include one-dimensional chains exhibiting parafermion zero modes [59], in two dimensions the topologically ordered string-net models [60] and in three dimensions the Walker-Wang models [61]. Interestingly, as we have shown in Ref. [50], the ground states of these families of models vary enormously in their complexity: while some of the topologically-ordered models can be expressed as free fermion states, others attain all possible values of interaction distance, including values that saturate the upper bound for $D_{\text{ent}}$ [Eq. (22)] in the thermodynamic limit.

Parafermion chains, parametrised by the group $\mathbb{Z}_N$, include the Majorana chain as a special case for $N = 2$. It is well known that Majorana chains can be expressed as free fermions models, so it comes as no surprise that they also have $D_{\text{th}}^{\beta} = 0$ and $D_{\text{ent}} = 0$. This characteristic made it possible to analytically study Majorana zero modes to a great extent as well as made them amenable to experimental implementations. On the other hand, parafermion models in general are strongly interacting. As a consequence, much less is theoretically known about their behaviour or how to realise them in the laboratory. By evaluating the interaction distance for these models at the fixed point, we analytically found that all models with $N = 2^n$, $n$ integer, have $D_{\text{ent}} = 0$ in their ground state [50], see Fig. 8. Hence, they can be related by local unitaries to $n$ copies of Majorana chains. This relation allows to determine the properties of the $\mathbb{Z}_{2^n}$ parafermions in terms of the known physics of Majorana fermions, such us their stability in terms of perturbations. Recently, several studies have focused on investigating the

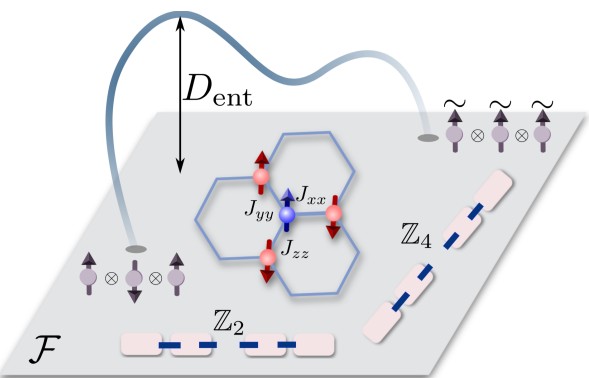

Figure 8: Different topological states, including the ground states of parafermion chains $\mathbb{Z}_2$ and $\mathbb{Z}_4$ as well as the Kitaev honeycomb lattice model, are interpreted as free-fermion states belonging to $\mathscr{F}$.

description of the $\mathbb{Z}_4$ model in terms of free fermions [62,63].

More surprisingly we observed that systems that support topological order, such as the toric code, have both thermal and entanglement interaction distances identically zero [50]. In fact, all Abelian $Z_{2^n}$ string-net models are unitarily equivalent to free fermions as well as their three-dimensional generalisations to the Walker-Wang models. This gives a new perspective to the origins of topological order and of anyonic statistics. In parallel, it makes these models amenable to a variety of analytical or numerical investigations that are hard to perform in strongly correlated systems beyond one dimension.

In conclusion, the interaction distance, as a diagnostic tool, has already yielded unique insights into a wide variety of interacting systems, ranging from standard many-body models (like Ising or Fermi-Hubbard) to more exotic models with topological order. We envision that the interaction distance can be embedded into a new class of numerical or analytical methods for solving interacting systems. For example, building a density functional theory that optimises over the entanglement properties rather than the local densities can bring about the optimal free model without the need to first determine the ground state of the model. Such an approach can be applicable not only in the perturbative regime where the Kohn-Sham model is known to be a good approximation, but more importantly in the strongly correlated regimes [56]. At the numerical front we envision that a variational method (akin to DMRG-type methods) can be employed that optimises with respect to $D_{\text{ent}}$. As the interaction distance is well behaved under renormalisation, confirmed by the system size scaling analysis [28], we are optimistic that it can reveal further surprises in the low energy behaviour of strongly interacting systems.

## 10 Acknowledgements

We would like to thank Irene D'Amico, Ashk Farjami, Konstantinos Meichanetzidis, Kristian Patrick and Christopher Turner for previous collaborations on the subject.

**Funding information** J.K.P. and Z.P. acknowledge support by EPSRC grants EP/I038683/1, EP/P009409/1, and EP/R020612/1.Statement of compliance with EPSRC policy framework on research data: This publication is theoretical work that does not require supporting research data.

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
