# Peer review of "Quantifying the effect of interactions in quantum many-body systems"

_SciPost Physics Lecture Notes, doi:SciPost Phys. Lect. Notes 4 (2018)_

## Round 1 · Referee Report · Anonymous (Referee 1) · 2018-5-28

Strengths

1- clear and accessible presentation of an new concept, the interaction distance, that affords new perspectives on quantum many-body systems
2- raises and suggests a number of questions about this measure that may spur further research based on the results presented

Weaknesses

1- the subject is very new and there is not yet a broad literature to review
2- the main new contents are fairly elementary (non-degenerate perturbation theory applied to D_th and detailed discussion of a 4-mode example)
3- the review of results obtained for large systems is very cursory
4- there is no discussion comparing D_F with other measures of interactivity (e.g. those arising from violation of Wick's theorem like non-factorizing higher moments).

Report

The authors review a concept they introduced recently that aims to
measure how close a given physical system (characterized by a state)
is to a free (Gaussian) system. Specifically, they study the
trace-distance of the systems density matrix from the set of free
states and consider the case of thermal states of a given system and
reduced density matrices arising from particular eigenstates.

The concept is motivated by the emergence of effectively free
quasi-particles and a number of general properties of the distance are
derived, chief among them that it is a purely *spectral* property and
its calculation can be reduced to a minimization over all possible
free spectra (given the eigenvalues of the interacting state) and a
global upper bound. A perturbative approach to computing the
interaction distance is then discussed and a simple example (the
four-mode "Hubbard dimer") discussed in detail, confirming the range
of applicability of the perturbative result and providing an example
where large (peaking) interaction distance coincides with a critical
point, while a distance approaching zero indicates that an effective
free model exists for very small and very large interactions.

The conclusions give an abbreviated summary of results obtained applying the
method to larger systems, including several examples of larger systems
to which the method has been applied.

The authors review an an interesting and very recent topic, providing a clear
introduction to the basic motivation and definition of the interaction
distance and illustrating its use and illustrating its use with a
transparent example. The presentation suggests a number of interesting
questions related to D_F and its properties and uses.

Given that the concept is very new (it was first proposed two years ago) and
has been employed, as far as I can see, up till now only in a handful of papers, it
may be a bit early to write a review since, in particular, Ref. [19]
does a good job in introducing the subject. The main new elements of
the present manuscript are the perturbation-theory analysis of
interaction distance (for the case of thermal states) and the detailed
presentation of the four-mode example.

According to the SciPost website, review papers are expected to
provide a "snapshot of a research area, written by recognized leaders
in the field, providing a critical assessment of current frontline
research and providing pointers towards future opportunities".
This is where I am most in doubt concerning the present manuscript: it
appears to review a still rather narrow subject (that probably cannot
yet be called a "research area") and I wouldn't call it a "critical
assessment" as it is not comparing the interaction-distance method" to
other approaches to detect/confirm effectively free behaviour.

Thus, in my opinion, the authors should consider to add more "meat" to
their otherwise nicely written review. For one, almost no contact is made to other
attempts to measure non-freeness like, for example, those based on
non-factorization of higher moments [of Fermi operators such as those
in eq.(15)] or quantitative deviations from mean-field
treatment or how previously discovered effective free models have been
optimized. Placing the new measure in that context would put its
characteristics and usefulness in stronger relief.
On the other hand, the summary of the results obtained by applying the
method to larger systems (Refs. [19,41,45]) in the concluding section is
very brief and I think that reviewing these results in more detail,
highlighting the qualitative insights as well as the practical
possibilities and limitations of the method would be useful.

Requested changes

1- add material placing the interaction-distance method in the context of prior studies of
effectively free models
2- provide a more detailed review of the applications of the method to larger systems, including
some plots and assessment of the conceptual and practical of the method: how useful is it if the
system is not exactly effectively free? can one already make statements for what kind of
interactions the method works well and why?

3- Now follow a number of smaller items which may require changes or comments from the
authors.
The set of free/Gaussian states (or the set of free spectra within R^{2^N}) is not convex;
maybe it would be advantageous to have that fact reflected in Fig. 2.
In general, lack of convexity may imply that there is no unique closest
free state/free spectrum to a given non-free state rho, while in the
authors seem to tacitly assume that there is (referring to "the
closest state" in several occasions). Could they comment on this issue?

4- For any given state rho there is a unique free state with the same
covariance matrix (the "Gaussian reference" of Ref. [38]) which is
in many respects the "free state closest to rho". Does it have any
relation to the one minimizing the interaction distance? (I assume
that not since the non-free transformation U will in general change
the covariance matrix of the state.)

5- The interaction distance is sometimes taken as a property of a state,
sometimes as that of a system, described by a Hamiltonian. One can
take both perspectives, but I think it would be worthwhile to
differentiate: in some cases, D_th(rho) may only allow statements
about that specific state (and its vicinity), and only
the knowledge of {D_th(rho(beta)) all beta>0} fully characterizes
the system, while in other cases (eg., if D_th(rho)=0 for a full-rank
state) it seems that a single state may allow to characterize the
system itself (i.e., the Hamiltonian).
For D_ent, it is not so clear if the results for a given eigenstate
have any bearing on other eigenstates or the system/Hamiltonian in
general.
I think that explaining these two perspectives (or explicitly
focusing on one of them) might further clarify the exposition.

6- Is there a particular reason to define interaction distance using the trace
distance (rather than other metrics on Hilbert space like the
Hilbert-Schmidt-distance/2-norm or relative entropy)?
What is the interaction distance of a random/generic state in a 2^N
dimensional Hilbert space? I think that number would be more telling than
the upper bound in what makes a state "close" to F.

7- p12: the authors point out that D_th and D_ent can each be zero when
the other is not. While this is clearly correct, it seems
counter-intuitive due to the fact that D_th=0 implies that the
system is described exactly by effective free quasi-particles (H of
form eq.(15), and, in particular, all eigenstates are free in this
new basis. Can this be used to argue that D_ent must be small,
since if the regions A, B are much larger than the size of the
quasiparticles, also the reductions of the eigenstates should be
free (up to contributions from quasiparticles living on the
boundary)?

8- p16, eq.(33): shouldn't there be a 2nd correction on the RHS: in addition
to the one arising from E_k = E_k^f + Delta E_k one from the perturbative
correction to Z? (which would replace |Delta E_k| by |Delta E_k-\sum_l Delta
E_l e^{-beta E^f_l}|)

9- p21: the authors characterize D_F as a "diagnostic tool" and state
that finding the spectrum and/or ground state is first needed. I
have several questions on this: how useful is the distance if only
the spectrum but not the corresponding eigenstates are known?
If only a part of the spectrum is known, say the first m<<2^N
eigenvalues, can one still obtain useful bounds on D_F? How large would m
have to be (at given temperature)?

10- Minor issues:

* p6: thermalization: the authors appear to imply that free
systems cannot thermalize. This seems to be in contradiction with
results such as PRL 117, 190602 (2016) or PRL 100, 030602 (2008)
for bosonic systems. Maybe the authors could clarify.

* p7, after eq.(11): better "beta E_k >> 1"

* p9, 1st line: due to the ambiguity of "non-linear" maybe add
"(non-free/non-Gaussian)" and/or that U is unitary.

* p10, eq.(17): maybe note that states that do not have full rank are only
limits of expressions such as eq.(17).

* p10: "we consider a generic density matrix": what means generic? which
"non-generic" effects are excluded here?

* p11, after eq.(19): in which sense is the computation "efficient"
given that there are exponentially many terms in the sum eq.(19)?

* p11, the measures for non-Gaussianity (of boson states) are nor
restricted to single modes. On might add QIP 11(3), 853–872 (2012),
which also introduces such a measure. The main difference to the
authors' approach is that these measures do not allow for non-linear
transformations of the mode operators.

* p12, Fig. 3, caption typo "compliment"

* p12: the authors write that to conclude that the correlations in a state
|Psi_k> are those of free fermions it is necessary (but enough) to check the
interaction distance of the reduced DM for all admissible bipartitions.
It is not clear to me that this is sufficient. How can one exclude the
possibility that all reduced states are Gaussian but there is no global
Gaussian state compatible with all reductions?

* p12: which bipartitions are to be considered? the discussion of
"short-distance" physics and "size of quasi-particles" suggests
spatially contiguous ranges. or should one also think about
bipartite lattices or partitions according to internal degrees of
freedom like spin?

* p13: the authors point out the flexibility in adding empty modes, i.e., that
in general D_F can be lowered by adding empty modes: D_F(rho\oplus 0_m) <=
D_F(rho). Are there examples where this makes a significant difference?
Should one then define the lim_{m\to\infty} as the true interaction
distance? Is there a finite m (fixed by the rank of the original
state?) beyond which it is not useful to go?

* p17: after eq.(45) the authors write about "the transition", later on they
mention "critical point" (p18) and "thermodynamic-limit transition point"
(p20), but in the initial discussion of the model no such point is
mentioned. I think that should be done before the interaction-distance
calculation and discussion.

* p20: "determining the interaction distance ... also specifies the optimal
free state": however, as far as I can see, knowing the distance
does, in general, not directly give the closest state - hence is
this conclusion specific to the computation of D_F via full
diagonalization of rho (that also gives their eigenstates)?

* p21: about the Fermi-Hubbard model and the parafermion chains at the
fixed point having zero D_ent: to which eigenstate(s) of the system
does D_ent refer here?

* references: Ref. [42] duplicates [14]; typos (missing capitalization of
German nouns resp. proper names in Refs. [8, 11, 25, 28, 30 43, 44, 50]); is
"z n" should probably Z_N in Ref. [46]? The doi in [7,28] should not
contain the URL.

  • validity: top
  • significance: good
  • originality: good
  • clarity: high
  • formatting: excellent
  • grammar: good

Anonymous on 2018-06-17  [id 275]

(in reply to Report 1 on 2018-05-28)

We thank the Referee for his/her detailed reading of our paper and for providing an extensive list of comments that helped us improve the presentation of our manuscript. The Referee wrote that we provided a “clear introduction” to an “interesting” and “very recent” topic. The Referee expressed some doubts over whether our manuscript meets the criteria for a “review article” in SciPost.

We emphasise that our manuscript was submitted to “Lecture Notes” category of SciPost, thus we did not aim to give a review of this research topic. (We apologise for not using the correct template for “Lecture Notes” in the TeX file, which would have shown this more clearly. We have now fixed this.) Therefore, our motivation was to provide an accessible introduction at the level of Master and PhD students. Indeed, as the Referee correctly observed, the subject of our manuscript is very new and the method has only been used in a few papers. Thus, we agree with the Referee that a standard review paper would be premature at this point. The main purpose of this manuscript is to inspire a broader investigation of the emerging topic we believe to be important, and to clearly introduce the required theoretical concepts at a pedagogical level that would be directly helpful to other researchers. Because of this, we have not performed the extensive further calculations that were suggested by the Referee:

>2- provide a more detailed review of the applications of the method to larger systems, including
>some plots and assessment of the conceptual and practical of the method: how useful is it if the
>system is not exactly effectively free? can one already make statements for what kind of
>interactions the method works well and why?

We believe these suggestions are valid and interesting to investigate, but they would change the scope of our manuscript into a research article, which is not our goal. However, we did take into account all the suggestions of the Referee how to improve the presentation and we have added citations to some of the suggested works in the literature. Below we address in detail the individual comments of the Referee.

>1- add material placing the interaction-distance method in the context of prior studies of
>effectively free models

As we do not aim to provide a review on the topic we restrict in these lecture notes to describe textbook methods, like the mean field theory and the density functional theory. These are presented and referenced in the introduction, page 1. For completeness, we now include 7 references on prior studies with a more specialised aim of quantifying Gaussianity of quantum states.

>3- The set of free/Gaussian states (or the set of free spectra within R^{2^N}) is not convex;
>maybe it would be advantageous to have that fact reflected in Fig. 2.
>In general, lack of convexity may imply that there is no unique closest
>free state/free spectrum to a given non-free state rho, while in the
>authors seem to tacitly assume that there is (referring to "the
>closest state" in several occasions). Could they comment on this issue?

We do not have a deep understanding of the structure of the manifold F. We agree with the Referee that in general there may not be a unique optimal free model. We have not directly encountered such a possibility in any of the cases we have studied so far. Thus, the cases with multiple optimal free models might be an exception rather than a general rule. Because of this, we have decided not to modify Fig.2, which is supposed to represent a general sketch rather than precise map of the manifold F. However, we have inserted a comment in the text about the possibility of the optimal free state not being unique.

>4- For any given state rho there is a unique free state with the same
>covariance matrix (the "Gaussian reference" of Ref. [38]) which is
>in many respects the "free state closest to rho". Does it have any
>relation to the one minimizing the interaction distance? (I assume
>that not since the non-free transformation U will in general change
>the covariance matrix of the state.)

We respectfully disagree with the Referee that the “Gaussian reference” state is “in many respects the free state closest to rho”. In previous works in the quantum information literature, such as Ref.38, the variational optimization over the free modes was not performed. Thus, the Gaussian reference state used there is practically never the optimal free state that we find. This can be seen, e.g., in the quantum Ising model, which we find to be free for any value of the transverse field.

>5- The interaction distance is sometimes taken as a property of a state,
>sometimes as that of a system, described by a Hamiltonian. One can
>take both perspectives, but I think it would be worthwhile to
>differentiate: in some cases, D_th(rho) may only allow statements
>about that specific state (and its vicinity), and only
>the knowledge of {D_th(rho(beta)) all beta>0} fully characterizes
>the system, while in other cases (eg., if D_th(rho)=0 for a full-rank
>state) it seems that a single state may allow to characterize the
>system itself (i.e., the Hamiltonian).
>For D_ent, it is not so clear if the results for a given eigenstate
>have any bearing on other eigenstates or the system/Hamiltonian in
>general.
>I think that explaining these two perspectives (or explicitly
>focusing on one of them) might further clarify the exposition.

The Referee’s intuition is correct; indeed, D_th and D_ent have different physical meaning, and different conclusions can be drawn by studying them separately. This is discussed at length in our Section 6.2, which carefully defines and distinguishes these two quantities. In particular, we caution that knowing D_ent for a single state provides limited information about the entire system, and also noted the dependence on the entanglement partition. We also discuss the interrelation between D_th and D_ent in the same section.

>6- Is there a particular reason to define interaction distance using the trace
>distance (rather than other metrics on Hilbert space like the
>Hilbert-Schmidt-distance/2-norm or relative entropy)?
>What is the interaction distance of a random/generic state in a 2^N
>dimensional Hilbert space? I think that number would be more telling than
>the upper bound in what makes a state "close" to F.

Indeed, it is not necessary to use trace distance, which is merely one convenient choice to define D_F. Other quantities, such as relative entropy, can equally be used. We have inserted a comment about this in the paper.

Regarding D_F of a random state, in unpublished work we have investigated this in some detail for the case of entanglement interaction distance. We were not able to obtain analytical result for D_ent of a random state, but from numerically sampling random states, we found that the probability distribution of D_ent is log-normal, with a peak between 0.01-0.02. This is indeed much smaller than the maximal value of D_ent, which is roughly 1/6. Since these are unpublished results, which require much further work to be analytically substantiated, we have decided not to include them in these lecture notes.

However, we are not sure whether we agree with the Referee that D_ent of a random state is “more telling” than the upper bound (1/6). Our focus is mainly on ground states of ordered systems, which are rather far from random states.
Random states are a good starting point for the discussion of thermal (highly excited) states, which have close-to-maximal entanglement. Entanglement is simply a very different concept from interaction distance. The maximally interacting states we identified with parafermion or string-net ground states are indeed very far from random states and of much interest for their fundamental properties and for their applications.

>7- p12: the authors point out that D_th and D_ent can each be zero when
>the other is not. While this is clearly correct, it seems
>counter-intuitive due to the fact that D_th=0 implies that the
>system is described exactly by effective free quasi-particles (H of
>form eq.(15), and, in particular, all eigenstates are free in this
>new basis. Can this be used to argue that D_ent must be small,
>since if the regions A, B are much larger than the size of the
>quasiparticles, also the reductions of the eigenstates should be
>free (up to contributions from quasiparticles living on the
>boundary)?

It may be common to find an agreement between the values of D_ent and D_th, when they become zero and when they are non-zero. Nevertheless, in strongly interacting cases that are of interest due to their exotic properties, such as string-net models or parafermion models, one could expect to establish such conditions independently from each other. We felt it is best to leave this general possibility open. Further investigations are needed to draw more concrete results about the relation between D_ent and D_th in specific cases.

>8- p16, eq.(33): shouldn't there be a 2nd correction on the RHS: in addition
>to the one arising from E_k = E_k^f + Delta E_k one from the perturbative
>correction to Z? (which would replace |Delta E_k| by |Delta E_k-\sum_l Delta
>E_l e^{-beta E^f_l}|)

We thank the Referee for pointing out this mistake. We have corrected the expression by substituting |Delta E_k| with |Delta E_k-\sum_l Delta E_l e^{-beta E^f_l}/Zf|. This error, however, has no bearing on the results on Section 8 where the correct expression was used in the first place.

>9- p21: the authors characterize D_F as a "diagnostic tool" and state
>that finding the spectrum and/or ground state is first needed. I
>have several questions on this: how useful is the distance if only
>the spectrum but not the corresponding eigenstates are known?
>If only a part of the spectrum is known, say the first m<<2^N
>eigenvalues, can one still obtain useful bounds on D_F? How large would m
>have to be (at given temperature)?

Indeed, we emphasize that D_F can be obtained efficiently if only the spectrum of a density matrix is known. While this in itself is an interesting possibility (which was not appreciated before our work), clearly there is much more information about the system which can be determined from the eigenvectors of the density matrix. For physical applications, such as in condensed matter physics, it is particularly important to know when D_F vanishes. In such regimes of parameters, other techniques can then be employed to find approximately free description of the system. For example, this is an approach we have used in classifying parafermion chains.

It would indeed be interesting to know what is the bound on D_F when only a part of the spectrum is known. Unfortunately, this is rather challenging, because D_F involves a non-linear optimization problem. We leave this question to future investigations.

>* p6: thermalization: the authors appear to imply that free
>systems cannot thermalize. This seems to be in contradiction with
>results such as PRL 117, 190602 (2016) or PRL 100, 030602 (2008)
>for bosonic systems. Maybe the authors could clarify.

We thank the Referee for raising this point. Thermalisation of free systems can occur locally, in subparts of the total Hilbert space corresponding to a conserved charge, as the indicated references clearly state. Nevertheless, this does not mean that a free system thermalises in its entirety. This is in contrast to what can happen in a generic interacting system. Our statement refers to the case of the entire closed system, and we do not see a conflict with the stated references.

>* p7, after eq.(11): better "beta E_k >> 1"

We have made the suggested modification.

>* p9, 1st line: due to the ambiguity of "non-linear" maybe add
"(non-free/non-Gaussian)" and/or that U is unitary.

We have added this clarification.

>* p10, eq.(17): maybe note that states that do not have full rank are only
limits of expressions such as eq.(17).

>* p10: "we consider a generic density matrix": what means generic? which
"non-generic" effects are excluded here?

We have replaced “generic” with “arbitrary”, which is what we had intended to say.

>* p11, after eq.(19): in which sense is the computation "efficient"
given that there are exponentially many terms in the sum eq.(19)?

Indeed, there are exponentially many (in the subsystem size) terms in the sum in Eq.19, but because they are exponentially decaying, the evaluation of the sum can be terminated once convergence is reached.

>* p11, the measures for non-Gaussianity (of boson states) are nor
restricted to single modes. On might add QIP 11(3), 853–872 (2012),
which also introduces such a measure. The main difference to the
authors' approach is that these measures do not allow for non-linear
transformations of the mode operators.

We thank the Referee for bringing to our attention the QIP reference, which we have now cited in the paper. Indeed, like other references on Gaussian quantum information that we cited, this paper is quite different from our work because it does not include the mentioned non-linear transformations of the mode operators.

>* p12, Fig. 3, caption typo "compliment"

We have fixed this typo.

>* p12: the authors write that to conclude that the correlations in a state
>|Psi_k> are those of free fermions it is necessary (but enough) to check the
>interaction distance of the reduced DM for all admissible bipartitions.
>It is not clear to me that this is sufficient. How can one exclude the
>possibility that all reduced states are Gaussian but there is no global
>Gaussian state compatible with all reductions?

We agree with the Referee that such a scenario is possible. To clarify it we added the following sentences: “In this case we assume that there is a global optimal Gaussian state that effectively describes the system. Nevertheless, it is possible that even if the correlations between all partitions are of free fermions, the total state is not reproducible by a Gaussian state.”

>* p12: which bipartitions are to be considered? the discussion of
>”short-distance" physics and "size of quasi-particles" suggests
>spatially contiguous ranges. or should one also think about
>bipartite lattices or partitions according to internal degrees of
>freedom like spin?

Because D_F is sensitive to correlations between the emerging quasiparticles, we consider contiguous real space partitions. We have inserted a comment about this in the text.

>* p13: the authors point out the flexibility in adding empty modes, i.e., that
>in general D_F can be lowered by adding empty modes: D_F(rho\oplus 0_m) <=
>D_F(rho). Are there examples where this makes a significant difference?
>Should one then define the lim_{m\to\infty} as the true interaction
>distance? Is there a finite m (fixed by the rank of the original
>state?) beyond which it is not useful to go?

Our optimisation algorithm for minimising D(rho-sigma) is constructed to add a number of such empty modes and verify if the trace distance becomes smaller. This we can verify numerically but we cannot make general statements as the optimisation is over non-linear functions.

>* p17: after eq.(45) the authors write about "the transition", later on they
>mention "critical point" (p18) and "thermodynamic-limit transition point"
>(p20), but in the initial discussion of the model no such point is
>mentioned. I think that should be done before the interaction-distance
>calculation and discussion.

We have inserted a comment motivating the transition in the Hubbard dimer, including a reference to its experimental realisation with cold atoms.

>* p20: "determining the interaction distance ... also specifies the optimal
>free state": however, as far as I can see, knowing the distance
>does, in general, not directly give the closest state - hence is
>this conclusion specific to the computation of D_F via full
>diagonalization of rho (that also gives their eigenstates)?

Yes, the guess of the Referee is correct. When one diagonalises rho then its eigenstates are available. The optimisation provides the eigenvalues of sigma in the same basis as rho. So the complete optimal state sigma is available by determining D_F.

>* p21: about the Fermi-Hubbard model and the parafermion chains at the
>fixed point having zero D_ent: to which eigenstate(s) of the system
>does D_ent refer here?

This refers to the ground state. We have clarified this in the text.

>* references: Ref. [42] duplicates [14]; typos (missing capitalization of
>German nouns resp. proper names in Refs. [8, 11, 25, 28, 30 43, 44, 50]); is
>”z n" should probably Z_N in Ref. [46]? The doi in [7,28] should not
>contain the URL.

We have cleaned up the formatting of all references.

---

## Round 3 · Referee Report · Anonymous (Referee 1) · 2018-6-29

Strengths

1- clear, up-to-date introduction to the topic 2- provides different, more accessible pace and style compared to research articles

Weaknesses

1- young and rather narrow topic, lecture notes may soon be superseded 2- method has not yet been adopted beyond its inventors

Report

The authors have addressed the points I raised and made a number of small changes to the manuscript, while leaving other questions to future research, which is fair enough as the lecture notes should
reflect the state of current knowledge. (I apologize for not having noticed the designation as "Lecture Notes": it was mentioned in the email but not noticeable from the manuscript.) ..............................................................................................................................................................................................................
I think the manuscript provides a clear and didactic introduction to
the interaction distance, summarizing the present state of knowledge. It
provides, in particular (and in distinction from the research papers
on which it is based), a broad introduction and motivation and a
detailed discussion of a simple application.
........................................................................................................................................................

As SciPost does not specify any restictions regarding the expected breadth or narrowness of the lecture notes, nor regarding the maturity of the topics covered by the same, and since the present manuscript is of good quality and covers subject matter belonging to the scope of SciPost Physics Lecture Notes, I recommend its publication.

Requested changes

0- not requested, only suggested changes, and answers to some comments by the authors. 1- regarding the "Gaussian reference state": I don't mean to argue with the authors about whether it is in "many" or "some" cases the free state closest to rho: it plays this role in the quantum information-related approaches the authors cite. Therefore, I think it would be useful and appropriate to point out that the authors' approach usually leads to a very different state.

2- regarding D_F of a random state: I don't insist on "more telling", but I would argue that D_F for a random state gives a better idea for what D_F should be considered "small" than only knowing D_F^max. I think that it would be worthwhile to include the authors' remarks on this subject is some form (footnote or remark) in the manuscript. I'm not sure how much of a role the "orderedness" of the system under consideration plays given that by definition interaction distance is optimized over an arbitrary unitary (that can upend all (e.g., local) structure possibly existing in the initial state). The question of what represents a random (unbiased) choice of a state brings its own complications and goes well beyond these lecture notes.

3- it might be worth remarking that the authors' approach (applied here to states) could equally well be discussed for Hamiltonians, unitaries, or completely positive maps - all of which also have a distinctive spectrum when they are free/Gaussian and for which the basis in which they closest to being free would be of interest.

  • validity: high
  • significance: good
  • originality: good
  • clarity: good
  • formatting: good
  • grammar: good

Author:  Zlatko Papic  on 2018-07-07  [id 288]

(in reply to Report 1 on 2018-06-29)

We thank the Referee for reviewing the changes we have made to the manuscript, and for recommending publication.

We have incorporated the final three comments of the Referee:

1 - At the end of Section 6.1, we note that minimisation in Eq.19 may lead to different sets of single particle energies $\{ \epsilon_i \}$ that give the same $D_F(\rho)$. We expect these coincidences to be rare and unstable against variations in the couplings of the model.

2 - In Section 6.3, we note that a random vector typically has $D_F$ much smaller than the upper bound.

3 - In Conclusions section, we note that analogous measure to $D_F$ can be generalised to compare unitaries or complete positive maps to their free fermion, Gaussian behaviour.

We hope that after these changes, our manuscript is ready for publication.

Sincerely,

The Authors

Author:  Zlatko Papic  on 2018-07-07  [id 287]

(in reply to Report 1 on 2018-06-29)

We thank the Referee for reviewing the changes we have made to the manuscript, and for recommending publication.

We have incorporated the final three comments on the Referee:

1 - At the end of Section 6.1, we note that minimisation in Eq.19 may lead to different sets of single particle energies $\{ \epsilon_i \}$ that give the same $D_F(\rho)$. We expect these coincidences to be rare and unstable against variations in the couplings of the model.

2 - In Section 6.3, we note that a random vector typically has $D_F$ much smaller than the upper bound.

3 - In Conclusions section, we note that analogous measure to $D_F$ can be generalised to compare unitaries or complete positive maps to their free fermion, Gaussian behaviour.

We hope that after these changes, our manuscript is ready for publication.

Sincerely,

The Authors

---

## Round 3 · Author Response

Dear Editor,

thank you for arranging the review of our manuscript and for forwarding us the report of the Referee.

We thank the Referee for his/her detailed reading of our paper and for providing an extensive list of comments that helped us improve the presentation of our manuscript. The Referee wrote that we provided a “clear introduction” to an “interesting” and “very recent” topic. The Referee expressed some doubts over whether our manuscript meets the criteria for a “review article” in SciPost.

We emphasise that our manuscript was submitted to “Lecture Notes” category of SciPost, thus we did not aim to give a review of this research topic. (We apologise for not using the correct template for “Lecture Notes” in the TeX file, which would have shown this more clearly. We have now fixed this.) Therefore, our motivation was to provide an accessible introduction at the level of Master and PhD students. Indeed, as the Referee correctly observed, the subject of our manuscript is very new and the method has only been used in a few papers. Thus, we agree with the Referee that a standard review paper would be premature at this point. The main purpose of this manuscript is to inspire a broader investigation of the emerging topic we believe to be important, and to clearly introduce the required theoretical concepts at a pedagogical level that would be directly helpful to other researchers. Because of this, we have not performed the extensive further calculations that were suggested by the Referee:

>2- provide a more detailed review of the applications of the method to larger systems, including
>some plots and assessment of the conceptual and practical of the method: how useful is it if the
>system is not exactly effectively free? can one already make statements for what kind of
>interactions the method works well and why?

We believe these suggestions are valid and interesting to investigate, but they would change the scope of our manuscript into a research article, which is not our goal.

However, we did take into account all the suggestions of the Referee how to improve the presentation and we have added citations to some of the suggested works in the literature. Below we address in detail the individual comments of the Referee.

>1- add material placing the interaction-distance method in the context of prior studies of
>effectively free models

As we do not aim to provide a review on the topic we restrict in these lecture notes to describe textbook methods, like the mean field theory and the density functional theory. These are presented and referenced in the introduction, page 1. For completeness, we now include 7 references on prior studies with a more specialised aim of quantifying Gaussianity of quantum states.

>3- The set of free/Gaussian states (or the set of free spectra within R^{2^N}) is not convex;
>maybe it would be advantageous to have that fact reflected in Fig. 2.
>In general, lack of convexity may imply that there is no unique closest
>free state/free spectrum to a given non-free state rho, while in the
>authors seem to tacitly assume that there is (referring to "the
>closest state" in several occasions). Could they comment on this issue?

We do not have a deep understanding of the structure of the manifold F. We agree with the Referee that in general there may not be a unique optimal free model. We have not directly encountered such a possibility in any of the cases we have studied so far. Thus, the cases with multiple optimal free models might be an exception rather than a general rule. Because of this, we have decided not to modify Fig.2, which is supposed to represent a general sketch rather than precise map of the manifold F. However, we have inserted a comment in the text about the possibility of the optimal free state not being unique.

>4- For any given state rho there is a unique free state with the same
>covariance matrix (the "Gaussian reference" of Ref. [38]) which is
>in many respects the "free state closest to rho". Does it have any
>relation to the one minimizing the interaction distance? (I assume
>that not since the non-free transformation U will in general change
>the covariance matrix of the state.)

We respectfully disagree with the Referee that the “Gaussian reference” state is “in many respects the free state closest to rho”. In previous works in the quantum information literature, such as Ref.38, the variational optimization over the free modes was not performed. Thus, the Gaussian reference state used there is practically never the optimal free state that we find. This can be seen, e.g., in the quantum Ising model, which we find to be free for any value of the transverse field.

>5- The interaction distance is sometimes taken as a property of a state,
>sometimes as that of a system, described by a Hamiltonian. One can
>take both perspectives, but I think it would be worthwhile to
>differentiate: in some cases, D_th(rho) may only allow statements
>about that specific state (and its vicinity), and only
>the knowledge of {D_th(rho(beta)) all beta>0} fully characterizes
>the system, while in other cases (eg., if D_th(rho)=0 for a full-rank
>state) it seems that a single state may allow to characterize the
>system itself (i.e., the Hamiltonian).
>For D_ent, it is not so clear if the results for a given eigenstate
>have any bearing on other eigenstates or the system/Hamiltonian in
>general.
>I think that explaining these two perspectives (or explicitly
>focusing on one of them) might further clarify the exposition.

The Referee’s intuition is correct; indeed, D_th and D_ent have different physical meaning, and different conclusions can be drawn by studying them separately. This is discussed at length in our Section 6.2, which carefully defines and distinguishes these two quantities. In particular, we caution that knowing D_ent for a single state provides limited information about the entire system, and also noted the dependence on the entanglement partition. We also discuss the interrelation between D_th and D_ent in the same section.

>6- Is there a particular reason to define interaction distance using the trace
>distance (rather than other metrics on Hilbert space like the
>Hilbert-Schmidt-distance/2-norm or relative entropy)?
>What is the interaction distance of a random/generic state in a 2^N
>dimensional Hilbert space? I think that number would be more telling than
>the upper bound in what makes a state "close" to F.

Indeed, it is not necessary to use trace distance, which is merely one convenient choice to define D_F. Other quantities, such as relative entropy, can equally be used. We have inserted a comment about this in the paper.

Regarding D_F of a random state, in unpublished work we have investigated this in some detail for the case of entanglement interaction distance. We were not able to obtain analytical result for D_ent of a random state, but from numerically sampling random states, we found that the probability distribution of D_ent is log-normal, with a peak between 0.01-0.02. This is indeed much smaller than the maximal value of D_ent, which is roughly 1/6. Since these are unpublished results, which require much further work to be analytically substantiated, we have decided not to include them in these lecture notes.

However, we are not sure whether we agree with the Referee that D_ent of a random state is “more telling” than the upper bound (1/6). Our focus is mainly on ground states of ordered systems, which are rather far from random states.
Random states are a good starting point for the discussion of thermal (highly excited) states, which have close-to-maximal entanglement. Entanglement is simply a very different concept from interaction distance. The maximally interacting states we identified with parafermion or string-net ground states are indeed very far from random states and of much interest for their fundamental properties and for their applications.

>7- p12: the authors point out that D_th and D_ent can each be zero when
>the other is not. While this is clearly correct, it seems
>counter-intuitive due to the fact that D_th=0 implies that the
>system is described exactly by effective free quasi-particles (H of
>form eq.(15), and, in particular, all eigenstates are free in this
>new basis. Can this be used to argue that D_ent must be small,
>since if the regions A, B are much larger than the size of the
>quasiparticles, also the reductions of the eigenstates should be
>free (up to contributions from quasiparticles living on the
>boundary)?

It may be common to find an agreement between the values of D_ent and D_th, when they become zero and when they are non-zero. Nevertheless, in strongly interacting cases that are of interest due to their exotic properties, such as string-net models or parafermion models, one could expect to establish such conditions independently from each other. We felt it is best to leave this general possibility open. Further investigations are needed to draw more concrete results about the relation between D_ent and D_th in specific cases.

>8- p16, eq.(33): shouldn't there be a 2nd correction on the RHS: in addition
>to the one arising from E_k = E_k^f + Delta E_k one from the perturbative
>correction to Z? (which would replace |Delta E_k| by |Delta E_k-\sum_l Delta
>E_l e^{-beta E^f_l}|)

We thank the Referee for pointing out this mistake. We have corrected the expression by substituting |Delta E_k| with |Delta E_k-\sum_l Delta E_l e^{-beta E^f_l}/Zf|. This error, however, has no bearing on the results on Section 8 where the correct expression was used in the first place.

>9- p21: the authors characterize D_F as a "diagnostic tool" and state
>that finding the spectrum and/or ground state is first needed. I
>have several questions on this: how useful is the distance if only
>the spectrum but not the corresponding eigenstates are known?
>If only a part of the spectrum is known, say the first m<<2^N
>eigenvalues, can one still obtain useful bounds on D_F? How large would m
>have to be (at given temperature)?

Indeed, we emphasize that D_F can be obtained efficiently if only the spectrum of a density matrix is known. While this in itself is an interesting possibility (which was not appreciated before our work), clearly there is much more information about the system which can be determined from the eigenvectors of the density matrix. For physical applications, such as in condensed matter physics, it is particularly important to know when D_F vanishes. In such regimes of parameters, other techniques can then be employed to find approximately free description of the system. For example, this is an approach we have used in classifying parafermion chains.

It would indeed be interesting to know what is the bound on D_F when only a part of the spectrum is known. Unfortunately, this is rather challenging, because D_F involves a non-linear optimization problem. We leave this question to future investigations.

>* p6: thermalization: the authors appear to imply that free
>systems cannot thermalize. This seems to be in contradiction with
>results such as PRL 117, 190602 (2016) or PRL 100, 030602 (2008)
>for bosonic systems. Maybe the authors could clarify.

We thank the Referee for raising this point. Thermalisation of free systems can occur locally, in subparts of the total Hilbert space corresponding to a conserved charge, as the indicated references clearly state. Nevertheless, this does not mean that a free system thermalises in its entirety. This is in contrast to what can happen in a generic interacting system. Our statement refers to the case of the entire closed system, and we do not see a conflict with the stated references.

>* p7, after eq.(11): better "beta E_k >> 1"

We have made the suggested modification.

>* p9, 1st line: due to the ambiguity of "non-linear" maybe add
"(non-free/non-Gaussian)" and/or that U is unitary.

We have added this clarification.

>* p10, eq.(17): maybe note that states that do not have full rank are only
limits of expressions such as eq.(17).

>* p10: "we consider a generic density matrix": what means generic? which
"non-generic" effects are excluded here?

We have replaced “generic” with “arbitrary”, which is what we had intended to say.

>* p11, after eq.(19): in which sense is the computation "efficient"
given that there are exponentially many terms in the sum eq.(19)?

Indeed, there are exponentially many (in the subsystem size) terms in the sum in Eq.19, but because they are exponentially decaying, the evaluation of the sum can be terminated once convergence is reached.

>* p11, the measures for non-Gaussianity (of boson states) are nor
restricted to single modes. On might add QIP 11(3), 853–872 (2012),
which also introduces such a measure. The main difference to the
authors' approach is that these measures do not allow for non-linear
transformations of the mode operators.

We thank the Referee for bringing to our attention the QIP reference, which we have now cited in the paper. Indeed, like other references on Gaussian quantum information that we cited, this paper is quite different from our work because it does not include the mentioned non-linear transformations of the mode operators.

>* p12, Fig. 3, caption typo "compliment"

We have fixed this typo.

>* p12: the authors write that to conclude that the correlations in a state
>|Psi_k> are those of free fermions it is necessary (but enough) to check the
>interaction distance of the reduced DM for all admissible bipartitions.
>It is not clear to me that this is sufficient. How can one exclude the
>possibility that all reduced states are Gaussian but there is no global
>Gaussian state compatible with all reductions?

We agree with the Referee that such a scenario is possible. To clarify it we added the following sentences: “In this case we assume that there is a global optimal Gaussian state that effectively describes the system. Nevertheless, it is possible that even if the correlations between all partitions are of free fermions, the total state is not reproducible by a Gaussian state.”

>* p12: which bipartitions are to be considered? the discussion of
>”short-distance" physics and "size of quasi-particles" suggests
>spatially contiguous ranges. or should one also think about
>bipartite lattices or partitions according to internal degrees of
>freedom like spin?

Because D_F is sensitive to correlations between the emerging quasiparticles, we consider contiguous real space partitions. We have inserted a comment about this in the text.

>* p13: the authors point out the flexibility in adding empty modes, i.e., that
>in general D_F can be lowered by adding empty modes: D_F(rho\oplus 0_m) <=
>D_F(rho). Are there examples where this makes a significant difference?
>Should one then define the lim_{m\to\infty} as the true interaction
>distance? Is there a finite m (fixed by the rank of the original
>state?) beyond which it is not useful to go?

Our optimisation algorithm for minimising D(rho-sigma) is constructed to add a number of such empty modes and verify if the trace distance becomes smaller. This we can verify numerically but we cannot make general statements as the optimisation is over non-linear functions.

>* p17: after eq.(45) the authors write about "the transition", later on they
>mention "critical point" (p18) and "thermodynamic-limit transition point"
>(p20), but in the initial discussion of the model no such point is
>mentioned. I think that should be done before the interaction-distance
>calculation and discussion.

We have inserted a comment motivating the transition in the Hubbard dimer, including a reference to its experimental realisation with cold atoms.

>* p20: "determining the interaction distance ... also specifies the optimal
>free state": however, as far as I can see, knowing the distance
>does, in general, not directly give the closest state - hence is
>this conclusion specific to the computation of D_F via full
>diagonalization of rho (that also gives their eigenstates)?

Yes, the guess of the Referee is correct. When one diagonalises rho then its eigenstates are available. The optimisation provides the eigenvalues of sigma in the same basis as rho. So the complete optimal state sigma is available by determining D_F.

>* p21: about the Fermi-Hubbard model and the parafermion chains at the
>fixed point having zero D_ent: to which eigenstate(s) of the system
>does D_ent refer here?

This refers to the ground state. We have clarified this in the text.

>* references: Ref. [42] duplicates [14]; typos (missing capitalization of
>German nouns resp. proper names in Refs. [8, 11, 25, 28, 30 43, 44, 50]); is
>”z n" should probably Z_N in Ref. [46]? The doi in [7,28] should not
>contain the URL.

We have cleaned up the formatting of all references.

We hope that the revised version of our manuscript is now ready for publication in SciPost.

Sincerely,

The Authors

---

## Round 3 · List of Changes

1. We have now included 7 references on prior studies with a more specialised aim of quantifying Gaussianity of quantum states.

  2. We have inserted a comment about the possibility of the optimal free state not being unique.

  3. We have inserted a comment about the possibility of using other distance measures other than the trace distance.

  4. We have fixed the mistake in the perturbative expression for $|Delta E_k|$.

  5. We have added the condition $\beta E_k \gg 1$, as suggested by the Referee.

  6. We have added the clarification "non-free/non-Gaussian" to remove ambiguity of "non-linear", as suggested by the Referee.

  7. We have explained what "efficient" means after Eq.19.

  8. We have added the suggested reference QIP 11(3), 853–872 (2012).

  9. We have fixed a typo in Fig.3 caption.

  10. We have added a comment regarding the existence (or possible lack of) a global Gaussian state valid for all bipartitions, as prompted by the Referee.

  11. We have inserted a comment about the type of partitioning we consider in evaluating $D_\text{ent}$.

  12. We have inserted a comment motivating the transition in the Hubbard dimer, including a reference to its experimental realisation with cold atoms.

  13. We have clarified that we refer to the ground state of the Hubbard dimer.

  14. We have cleaned up the formatting of all references.

---

## Editorial Decision

published